# Prediction of primary somatosensory neuron activity during active tactile exploration

Dario Campagner[1], Mathew Hywel Evans[1], Michael Ross Bale[1,2], Andrew Erskine[1,3], Rasmus Strange Petersen[1]*

[1]Faculty of Life Sciences, The University of Manchester, Manchester, United Kingdom; [2]School of Life Sciences, University of Sussex, Brighton, United Kingdom; [3]Mill Hill Laboratory, The Francis Crick Institute, London, United Kingdom

**Abstract** Primary sensory neurons form the interface between world and brain. Their function is well-understood during passive stimulation but, under natural behaving conditions, sense organs are under active, motor control. In an attempt to predict primary neuron firing under natural conditions of sensorimotor integration, we recorded from primary mechanosensory neurons of awake, head-fixed mice as they explored a pole with their whiskers, and simultaneously measured both whisker motion and forces with high-speed videography. Using Generalised Linear Models, we found that primary neuron responses were poorly predicted by whisker angle, but well-predicted by rotational forces acting on the whisker: both during touch and free-air whisker motion. These results are in apparent contrast to previous studies of passive stimulation, but could be reconciled by differences in the kinematics-force relationship between active and passive conditions. Thus, simple statistical models can predict rich neural activity elicited by natural, exploratory behaviour involving active movement of sense organs.

*For correspondence:
R.Petersen@manchester.ac.uk

Competing interests: The authors declare that no competing interests exist.

## Introduction

A major challenge of sensory neuroscience is to understand the encoding properties of neurons to the point that their spiking activity can be predicted in the awake animal, during natural behaviour. However, accurate prediction is difficult without experimental control of stimulus parameters and, despite early studies of awake, behaving animals (*Hubel, 1959*), subsequent work has most often effected experimental control by employing anaesthesia and/or passive stimulation. However, the active character of sensation (*Gibson, 1962*; *Yarbus, 1967*), based on motor control of the sense organs, is lost in reduced preparations. Recent methodological advances permit a way forward: in the whisker system, it is now possible to record neuronal activity from an awake mouse, actively exploring the environment with its whiskers, whilst simultaneously measuring the fundamental sensory variables (whisker kinematics and mechanics) likely to influence neuronal activity (*O'Connor et al., 2010b*).

Our aim here was to predict spikes fired by primary whisker neurons (PWNs) of awake mice engaged in natural, object exploration behaviour. The manner in which primary neurons encode sensory information fundamentally constrains all downstream neural processing (*Lettvin et al., 1959*). PWNs innervate mechanoreceptors located in the whisker follicles (*Zucker and Welker, 1969*; *Rice et al., 1986*). They are both functionally and morphologically diverse; including types responsive to whisker-object contact and/or whisker self-motion (*Szwed et al., 2003*; *Ebara et al., 2002*). PWNs project to the cerebral cortex, analogously to other modalities, via trisynaptic pathways through the brainstem and thalamus (*Diamond et al., 2008*).

**eLife digest** The brain receives information from the world through the senses. In particular, cells called sensory neurons can detect signals from the environment and relay the information to the brain. A critical test of how well we understand the role of a given sensory neuron is whether it is possible to predict its activity under natural conditions. Previous research has succeeded in predicting the responses of sensory neurons in animals that were anaesthetised. However, it has been difficult to extend this approach to awake animals.

Mice and other rodents rely on their whiskers to tell them about their surroundings. Campagner et al. set out to predict how the sensory neurons that send information from whiskers (or 'whisker neurons') to the brain would respond in awake mice that were actively exploring an object in their environment. The approach involved using high-speed video (1,000 frames per second) to film the whiskers while the mice used them to explore a thin metal pole. At the same time, Campagner et al. recorded the electrical activity of the whisker neurons. The videos were used to calculate the forces acting on the whiskers, and then computational models were used to relate the activity of the neurons to the forces.

This approach allowed Campagner et al. to predict the responses of the whisker neurons, even when the mice were exploring the pole freely and unpredictably, simply from knowledge of the forces that were acting on the whiskers.

Together, these findings move the field of neuroscience forward by showing that sensory signals and neuronal responses can be correlated even in an awake animal. A key challenge for the future will be to further extend the approach to investigate how the signal conveyed by sensory neurons is transformed by neural circuits within the brain.

Here, we show that PWN responses are well-predicted by rotational force ('moment') acting on the whisker, while whisker angle is a poor predictor. Moment coding accounts for spiking during both whisker-object interaction and whisker motion in air. Moment coding can also account for findings in previous studies of passive stimulation in the anaesthetized animal; indicating that the same biomechanical framework can account for primary somatosensory neuron responses across diverse states. Our results provide a mechanical basis for linking receptor mechanisms to tactile behaviour.

## Results

### Primary whisker neuron activity during object exploration is predicted by whisker bending moment

We recorded the activity of single PWNs from awake mice (*Figure 1A,E*, *Figure 1—figure supplement 1*) as they actively explored a metal pole with their whiskers (N = 20 units). At the same time, we recorded whisker motion and whisker shape using high-speed videography (1000 frames/s; *Figure 1D*, *Video 1*). As detailed below, PWNs were diverse, with some responding only to touch, others also to whisker motion. Since each PWN innervates a single whisker follicle, we tracked the 'principal whisker' of each recorded unit from frame to frame, and extracted both the angle and curvature of the principal whisker in each video frame (total 1,496,033 frames; *Figure 1B–E*; *Bale et al., 2015*). Whiskers are intrinsically curved, and the bending moment on a whisker is proportional to how much this curvature changes due to object contact (*Birdwell et al., 2007*): we therefore used 'curvature change' as a proxy for bending moment (*O'Connor et al., 2010a*). Whisker-pole contacts caused substantial whisker bending (curvature change), partially correlated with the whisker angle (*Figures 1E*, *4E*) and, consistent with *Szwed et al. (2003)* and *Leiser and Moxon (2007)*, robust spiking (*Figures 1E*, *2E*).

To test between candidate encoding variables, our strategy was to determine how accurately it was possible to predict PWN activity from either the angular position or curvature change of each recorded unit's principal whisker. To predict spikes from whisker state, we used Generalised Linear Models (GLMs; *Figure 2A*). GLMs, driven by whisker angle, have previously been shown to provide a simple but accurate description of the response of PWNs to passive stimulation (*Bale et al., 2013*)

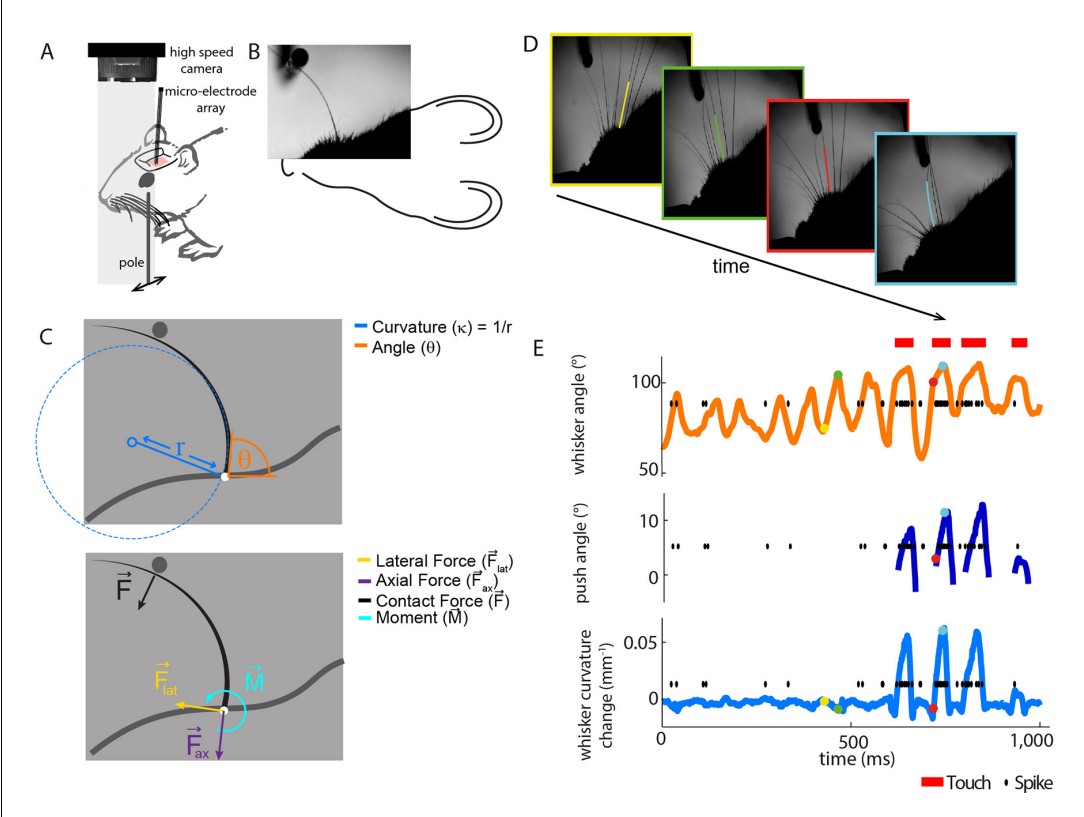

**Figure 1.** Electrophysiological recording from single primary whisker units in awake, head-fixed mice and simultaneous measurement of whisker kinematics/mechanics. (A) Schematic of the preparation, showing a tungsten microelectrode array implanted into the trigeminal ganglion of a head-fixed mouse, whilst a metal pole is presented in one of a range of locations (arrows). Before the start of each trial, the pole was moved to a randomly selected, rostro-caudal location. During this time, the whiskers were out of range of the pole. At the start of the trial, the pole was rapidly raised into the whisker field, leading to whisker-pole touch. Whisker movement and whisker-pole interactions were filmed with a high-speed camera. (B, C) Kinematic (whisker angle θ) and mechanical (whisker curvature κ, moment $\vec{M}$, axial force $\vec{F}_{ax}$ and lateral force $\vec{F}_{lat}$) variables were measured for the principal whisker in each video frame. When a whisker pushes against an object during protraction (as in panel D, red and cyan frames), curvature increases; when it pushes against an object during retraction (as in panels B and C), it decreases. (D) Individual video frames during free whisking (yellow and green) and whisker-pole touch (red and cyan) with tracker solutions for the target whisker (the principal whisker for the recorded unit, panel E) superimposed (coloured curve segments). (E) Time series of whisker angle, push angle and curvature change, together with simultaneously recorded spikes (black dots) and periods of whisker-pole contact (red bars). Coloured dots indicate times of correspondingly coloured frames in D.

The following figure supplements are available for figure 1:

**Figure supplement 1.** Electrophysiological recording from trigeminal primary neurons of awake, head-fixed mice.

**Figure supplement 2.** Computation of axial and lateral contact forces.

and have mathematical properties ideal for robust parameter-fitting (*Truccolo et al., 2005*; *Paninski et al., 2007*).

For each recorded unit (median 69,672 frames and 550 spikes per unit), we computed the GLM parameters that best predicted the unit's spike train given the whisker angle time series, using half the data as a training set for parameter-fitting (8 total fitted parameters - 5 for stimulus filter, 2 for history filter, 1 bias; *Figure 2—figure supplement 3*). We then assessed prediction performance using the other half of the data as a testing set: we provided the GLM with the whisker angle time series as input and calculated the predicted spike train, evoked in response (Materials and methods). We then compared the recorded spike train to the GLM-predicted one (*Figure 2B–C*) and quantified the similarity between the smoothed spike trains using the Pearson correlation coefficient (PCC). This is a stringent, single-trial measure of model prediction performance (*Figure 2—figure*

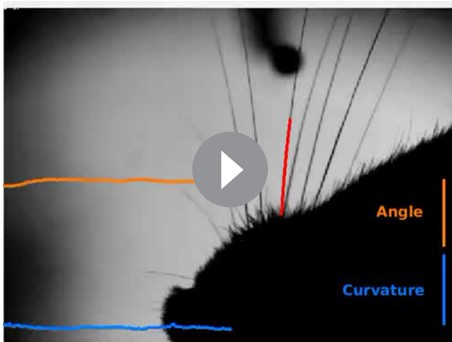

**Video 1.** Video of an awake mouse, exploring a pole with its whiskers with simultaneous electrophysiological recording of a primary whisker neuron.     At the start of the video, the pole is out of range of the whiskers. The whisker tracker solution for the principal whisker of the recorded unit is overlaid in red. White dots represent spikes; orange trace shows whisker angle (scale bar = 40°); blue trace shows whisker curvature change (scale bar = 0.05 mm$^{-1}$). Video was captured at 1000 frames/s and is played back at 50 frames/s. Related to *Figure 1*.

*supplement 1B*). We then repeated this entire procedure for the whisker curvature time series. Although angle GLMs predicted spike trains of a few units moderately well (2/20 units had PCC > 0.5), they performed poorly for the majority (median PCC 0.06, IQR 0.019–0.3; *Figure 2B–D*, orange). This was unlikely to be because of non-linear tuning to whisker angle, since quadratic GLMs fared only marginally better (median PCC 0.097, IQR 0.042–0.31; p=0.044, signed-rank test, *Figure 2—figure supplement 1A*). In contrast, we found that, at the population level, the curvature GLMs were substantially more accurate than the angle GLMs (median PCC 0.52, IQR 0.22–0.66; p=0.0044, signed-rank test; *Figure 2B–D*, blue) with prediction accuracy up to PCC 0.88. Curvature GLMs also predicted spikes during touch episodes significantly more accurately (median PCC 0.57, IQR 0.23–0.72) than did angle GLMs during non-touch episodes (median 0.06, IQR 0.02–0.35; p=0.005, signed-rank test). At the level of individual units, 90% had above chance PCC and we termed these 'curvature-sensitive' (Materials and methods). Of the curvature-sensitive units, 61% were sensitive to positive curvature change and 39% to negative curvature change (Materials and methods).

The result that curvature predicted PWN responses better than angle was robust to the number of fitted parameters: a GLM sensitive to instantaneous curvature (4 parameters: 1 stimulus filter parameter, 2 history filter parameters and 1 bias) exhibited very similar prediction accuracy (*Figure 2—figure supplement 1C*). The result was also robust to time-scale: prediction accuracy based on curvature was significantly greater than that based on angle for smoothing time-scales in the range 1–100 ms (signed-rank test, p<0.05, Bonferroni-corrected).

Although the activity of most units was better predicted by whisker curvature change than by whisker angle, there was significant variability in prediction performance, and there were a few units for which the angle prediction performance was appreciable (*Figure 2D*). However, we found that this could largely be attributed to redundancy. When a mouse whisks against an object, curvature change and angle fluctuate in concert (*Birdwell et al., 2007*; *Bagdasarian et al., 2013*; *Pammer et al., 2013*; *Figures 1E*, *4E* and *Figure 4F–G*). When we fitted GLMs using both curvature change and angle as input, these GLMs predicted the spike trains no more accurately (median PCC 0.53, IQR 0.40–0.62; p=0.067, signed-rank test; *Figure 2D*) than GLMs based on curvature change alone. Moreover, on a unit-by-unit basis, for 65% of units, curvature change GLMs predicted spikes better than angle (signed-rank test, p<0.05, Bonferroni-corrected); only for 5% of units did angle predict spikes better than curvature change. GLMs based on curvature change also predicted spike trains more accurately than GLMs based on 'push angle' – the change in angle as the whisker pushes against an object (*Figure 1E*; median PCC 0.25, IQR 0.04–0.45; p=0.006, signed-rank test). Moreover, prediction accuracy of GLMs fitted with both push angle and curvature change (median PCC 0.52, IQR 0.2–0.69) inputs was no better than that of GLMs fitted with curvature alone (p=0.43, signed-rank test).

In principle, neurons might also be sensitive to the axial force component (parallel to the whisker follicle) and/or lateral force component (orthogonal to axial) associated with whisker-object contact (*Figure 1B–C*, *Figure 1—figure supplement 2*; *Solomon and Hartmann, 2006*; *Pammer et al., 2013*). We restricted our analysis to bending moment since, under our experimental conditions, axial/lateral force components were near-perfectly correlated with bending moment (*Figure 2—figure supplement 2*) and bending moment is likely to have a major influence on stresses in the follicle (*Pammer et al., 2013*).

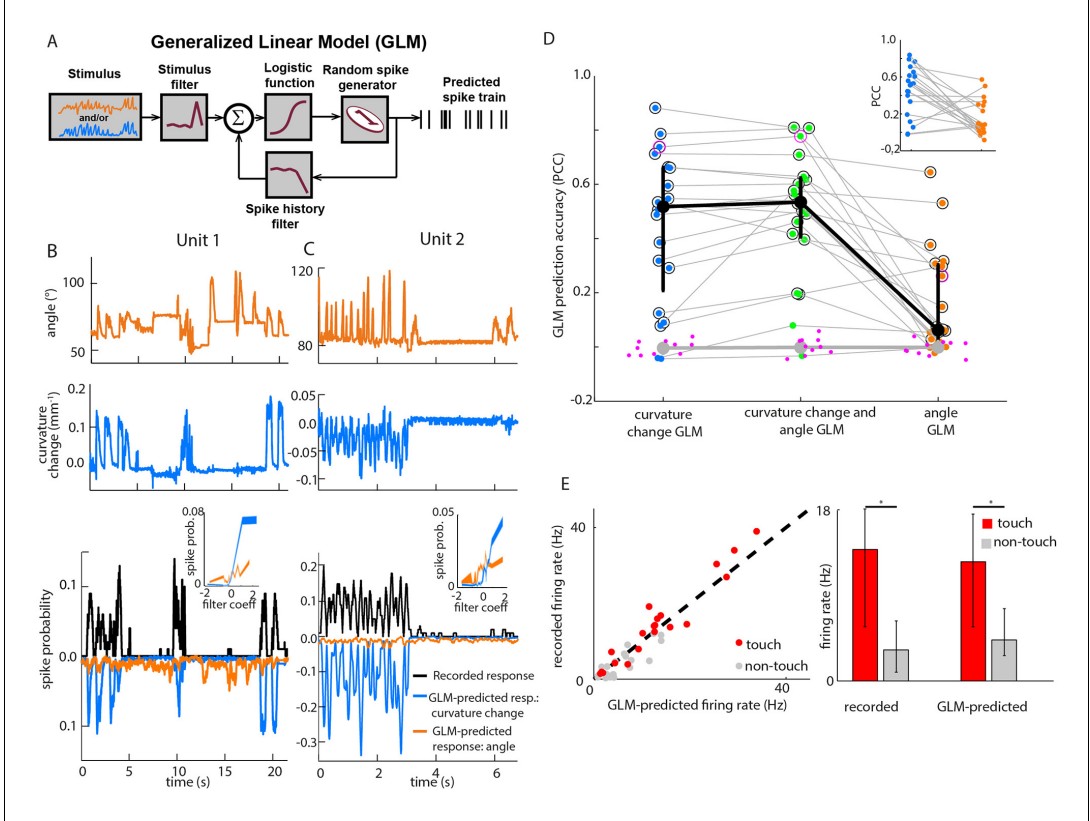

**Figure 2.** Primary whisker neurons encode whisker curvature, not whisker angle, during active sensation. (**A**) Schematic of the Generalized Linear Model (GLM). (**B**) For an example unit, whisker angle (top panel), whisker curvature change (middle panel) and simultaneously recorded spike train (bottom panel, black), together with predicted spike trains for the best-fitting angle GLM (bottom panel, orange) and curvature change GLM (bottom panel, blue). Spike trains discretized using 1-ms bins and smoothed with a 100 ms boxcar filter. Prediction performance (Pearson correlation coefficient, PCC) for this unit was 0.59. Inset shows tuning curves for both GLMs, computed by convolving the relevant sensory time series (angle or curvature change) with the corresponding GLM stimulus filter to produce a time series of filter coefficients, and estimating the spiking probability as a function of filter coefficient (25 bins). (**C**) Analogous to panel **B**, for a second example unit. Prediction performance PCC for this unit was 0.74. (**D**) Prediction performance (PCC between predicted and recorded spike trains) compared for GLMs fitted with three different types of input: curvature change alone; angle alone; both curvature change and angle. Each blue/orange/green dot is the corresponding PCC for one unit: large black dots indicate median; error bars denote inter-quartile range (IQR). To test statistical significance of each unit's PCC, the GLM fitting procedure was repeated 10 times on spike trains subjected each time to a random time shift: magenta dots show these chance PCCs for the unit indicated by the magenta circle; the mean chance PCC was computed for each unit and the large grey dot shows the median across units. Black circles indicate units whose PCC was significantly different to chance (signed-rank test, Bonferroni-corrected, p<0.0025). To facilitate direct comparison between results for curvature change GLM and angle GLM, these are re-plotted in the inset. (**E**) Left. Firing rate during touch episodes compared to that during non-touch episodes for each unit, compared to corresponding predicted firing rates from each unit's curvature change GLM. Right. Medians across units: error bars denote IQR; * denotes differences significant at p<0.05 (signed-rank test).

The following figure supplements are available for figure 2:

**Figure supplement 1.** Effect on GLM performance of quadratic input terms, simulated repeated trials and minimal stimulus filters.

**Figure supplement 2.** Moment is near-perfectly correlated with axial/lateral contact force components during pole exploration.

**Figure supplement 3.** Example filters for curvature-based GLMs.

To further test the curvature-encoding concept, we asked whether curvature GLMs could account for the response of PWNs to whisker-pole touch. To this end, we parsed the video data into episodes of 'touch' and 'non-touch'. Units fired at a higher rate during touch than otherwise (*Szwed et al., 2003*; *Leiser and Moxon, 2007*). Without any further parameter-adjustment, the curvature-based GLMs reproduced this effect (*Figure 2E*): the correlation coefficient between recorded

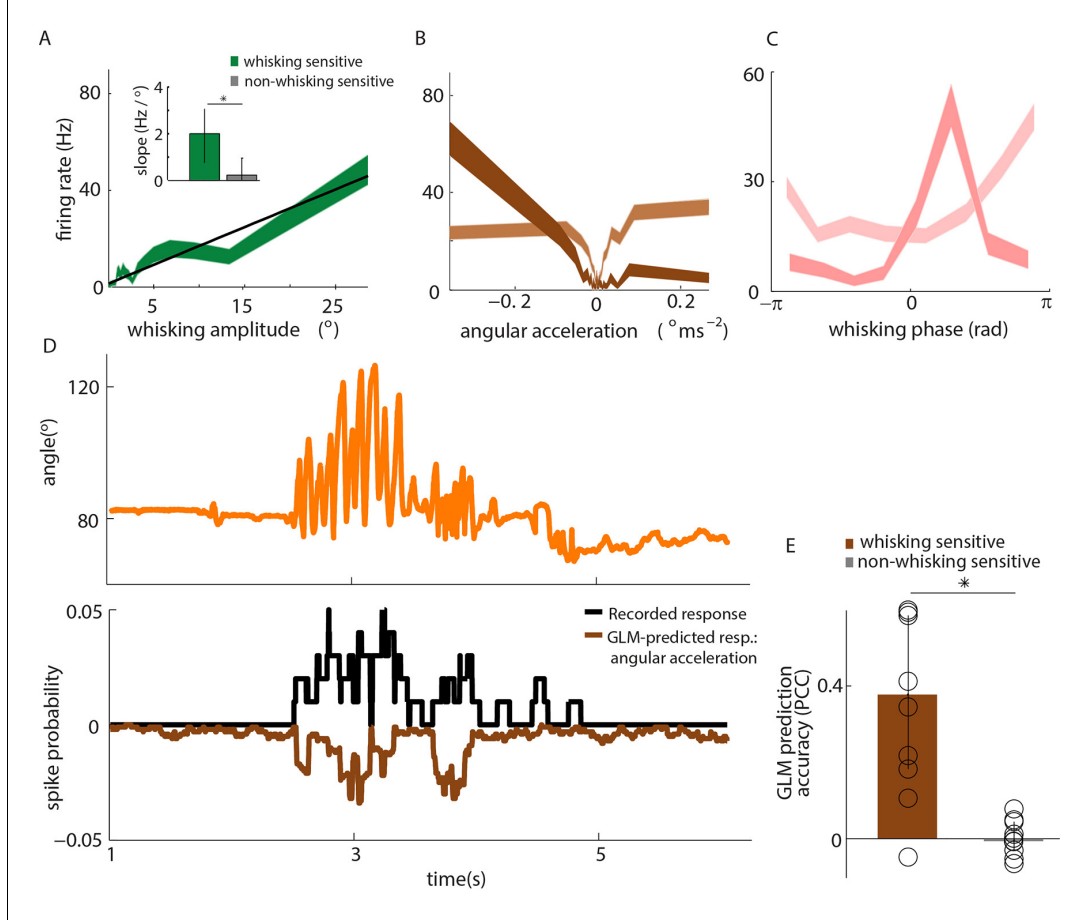

**Figure 3.** Primary whisker neurons encode whisker angular acceleration during free whisking. (A) Mean response of an example whisking-sensitive unit to whisking amplitude, computed during non-contact episodes (dark green, shaded area shows SEM) with regression line (black). Inset shows regression line slopes (median and IQR) for whisking sensitive (green) and whisking insensitive (grey) units. * indicates statistically significant rank-sum test (p=0.05). (B) Mean response of two example units as a function of angular acceleration. The dark brown unit is the same as that shown in A. (C) Mean response of two example units as a function of whisking phase. The dark pink unit is the same as that reported in A; the light pink unit is the same as that shown as light brown in B. (D) Excerpt of free whisking (orange) along with activity of an example, whisking-sensitive unit (black) and activity predicted by a GLM driven by angular acceleration (brown). The unit is the same as that shown in A. (E) GLM prediction accuracy (PCC) for all whisking sensitive (brown) and whisking insensitive units (grey). Bars and vertical lines denote median and IQR respectively.

The following figure supplement is available for figure 3:

**Figure supplement 1.** Whisking-sensitive units exhibit heterogeneous selectivity to angular acceleration.

and GLM-predicted firing rate for touch episodes was 0.97. Collectively, the above results indicate that, during active touch, the best predictor of whisker primary afferent firing is not whisker angle but rather the bending moment.

## Primary whisker neuronal activity during whisking is predicted by moment

During free whisking - in the absence of whisker-pole contact - whisker curvature, and therefore bending moment, changed little (*Figure 1E*, *Figure 4F*); consistent with previous studies (*Knutsen et al., 2008*; *Quist et al., 2014*). Yet, 50% of recorded units ('whisking-sensitive units') were significantly modulated by whisking amplitude (*Figure 3A*). Consistent with *Szwed et al. (2003)*, PWNs were diverse: 45% were curvature-sensitive (significant PCC for curvature based GLM) but not whisking-sensitive; 45% were both curvature- and whisking-sensitive; 5% were whisking-sensitive but not curvature-sensitive.

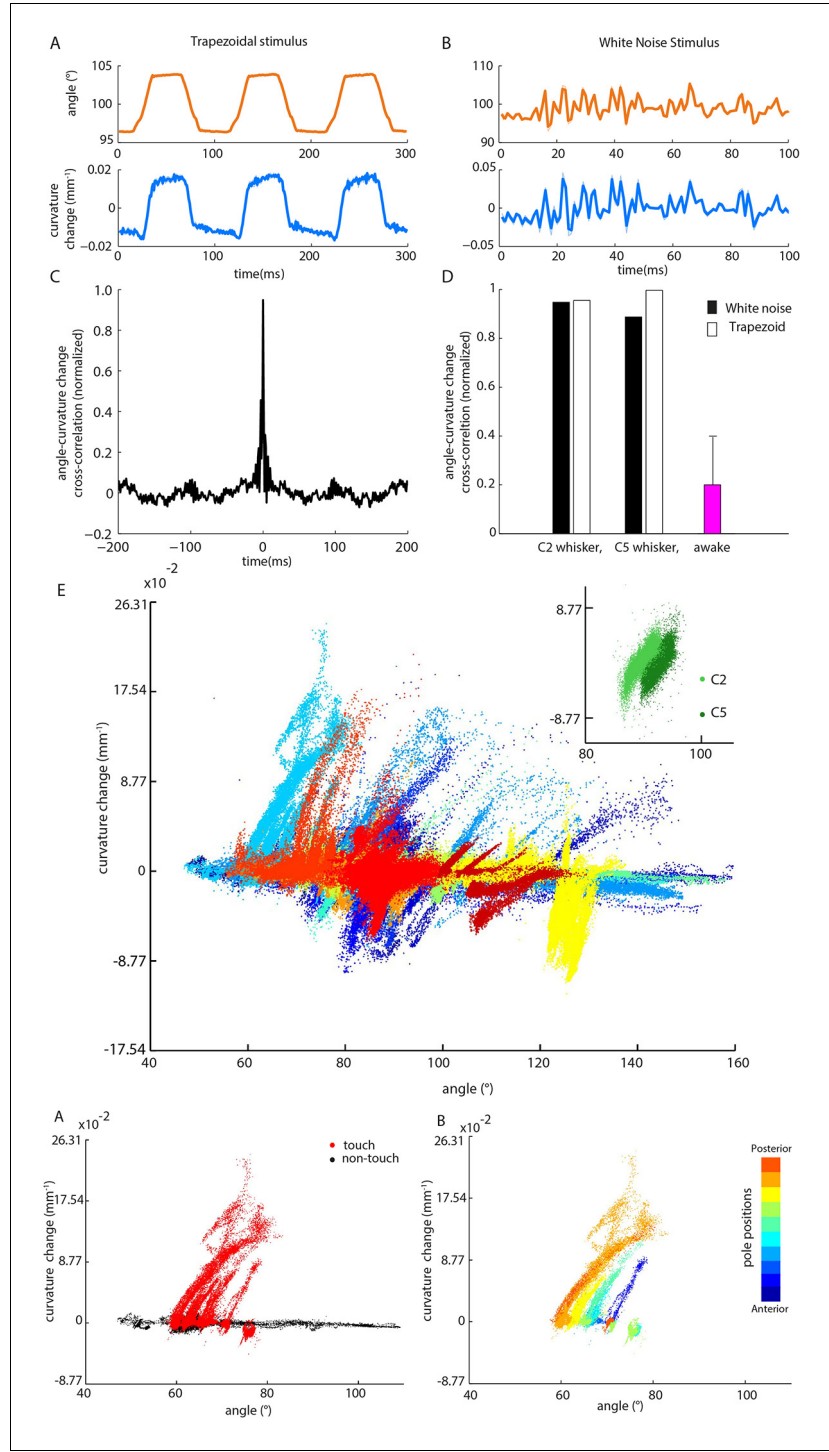

**Figure 4.** Whisker angle and whisker curvature change are highly correlated during passive whisker deflection, but decoupled during active touch. (**A**) Whisker angle (*top*) and whisker curvature change (*bottom*) time series, due to passive, trapezoidal stimulation of C2 whisker in an anaesthetized mouse, estimated as mean over 10 repetitions. Note that error bars (showing SEM) are present but very small. (**B**) Corresponding data for low-pass filtered white noise (hereafter abbreviated to 'white noise') stimulation of the same whisker. (**C**) Cross-correlation between curvature change and angle during white noise stimulation, for C2 whisker. (**D**) Cross-correlation between angle and curvature change at zero lag, for both passive stimulation under anaesthesia and awake, active sensing (median of absolute cross-correlation for each unit; error bar denotes IQR). (**E**) Joint distribution of whisker angle and whisker curvature change in awake, behaving mice (1 ms sampling). Different colours denote data

*Figure 4 continued on next page*

*Figure 4 continued*

corresponding to different recorded units. *Inset*: Analogous plot for passive, white noise whisker deflection in an anaesthetised mouse. Different colours indicate data from different whiskers. (F) Joint distribution of angle and curvature change for an example recording from an awake behaving mouse, with samples registered during touch and non-touch distinguished by colour (1 ms sampling). (G) Touch data of F classified according to pole position (dot colour).

The following figure supplements are available for figure 4:

**Figure supplement 1.** Correlations between angle and curvature change during passive whisker stimulation can make curvature-tuned units appear angle-tuned.
**Figure supplement 2.** Measurement of whisker bending during passive whisker deflection.

The presence of whisking sensitivity suggests that moment due to whisker bending is not the only force that influences PWN activity. A likely candidate is the moment associated with the rotational acceleration of a whisker: this moment is proportional to the whisker's angular acceleration (*Quist et al., 2014*; Materials and methods). Consistent with this possibility, we found that whisking-sensitive units were tuned to angular acceleration (*Figure 3B*) and that 50% of these were phase-modulated (*Figure 3C*). Angular acceleration tuning was diverse: some units fired to acceleration in a particular direction (rostral or caudal), whilst others responded to acceleration in both directions (*Figure 3B*, *Figure 3—figure supplement 1*). Moreover, for whisking-sensitive units (but not whisking-insensitive ones), quadratic GLMs trained on data from non-touch episodes were able to predict spikes using whisker angle acceleration as input (*Figure 3D–E*; whisking-sensitive units, median PCC 0.37, IQR 0.18–0.58; whisking-insensitive, median PCC -0.0071, IQR -0.035–0.041; p=0.0017 rank-sum test for whisking-sensitive vs whisking-insensitive units). For 70% of whisking-sensitive units, directional selectivity for acceleration was consistent with that for curvature. These findings indicate that, in the absence of whisker-object contact, responses of PWNs to whisking itself can be accounted for by sensitivity to the moment associated with angular whisker acceleration.

## Relation between kinematics and mechanics is different in active vs passive touch and has implications for neural encoding

We found, during active object exploration, that curvature change, but not whisker angle, predicts PWN firing. In apparent contrast, studies using passive whisker stimulation have reported that PWNs encode whisker angle and its temporal derivatives (*Zucker and Welker, 1969*; *Gibson and Welker, 1983*; *Lichtenstein et al., 1990*; *Jones et al., 2004*; *Arabzadeh et al., 2005*; *Bale and Petersen, 2009*; *Lottem and Azouz, 2011*; *Bale et al., 2013*). We wondered whether the discrepancy might be due to differences in whisker mechanics between passive and active stimulation conditions. To test this, we analysed the relationship between angle and curvature change during active touch and compared it to that during passive whisker stimulation. During active pole exploration, angle and curvature change were, over all, only loosely related (median correlation coefficient 0.20, IQR 0.079–0.39, *Figures 4D–E*). Important contributory factors were that the angle-curvature relationship was both different for touch compared to non-touch (*Figure 4F*) and dependent on object location (*Figure 4G*). In contrast, during passive stimulation, whisker angle was near perfectly correlated with curvature change (for C2, correlation coefficients 0.96 and 0.94 respectively; similar results for C5; *Figures 4C–D*, *Figure 4E*, inset and *Figure 4—figure supplement 2*); consistent with properties of cantilevered beams (*Birdwell et al., 2007*). Simulations confirmed that, due to the tight relationship between the variables, a unit tuned purely to curvature change can appear tightly tuned to angle (*Figure 4—figure supplement 1*). The implication is that apparent sensitivity to whisker angle under passive stimulation conditions can be accounted for by moment-tuning.

## Discussion

### Prediction of spikes fired by sensory neurons under natural conditions

In the endeavour to understand how neurons encode and process sensory information, there is a basic tension between the desire for tight experimental control and the desire to study animals under natural, unconstrained conditions. Theories of sensory encoding suggest that neural circuits have evolved to operate efficiently under natural conditions (*Simoncelli and Olshausen, 2001*; *Reinagel, 2001*). Previous studies have succeeded in predicting/decoding spikes evoked by passive presentation of natural sensory stimuli to anaesthetised/immobilised animals (*Lewen et al., 2001*; *Arabzadeh et al., 2005*; *Pillow et al., 2008*; *Mante et al., 2008*; *Lottem and Azouz, 2011*; *Bale et al., 2013*), but it has been difficult to extend this approach to encompass natural, active movement of the sense organs. Here, we have addressed this general issue, taking advantage of experimental possibilities recently created in the whisker system (*O'Connor et al., 2010a*), and the ability of computational methods, such as GLMs, to uncover stimulus-response relationships even from data with complex statistical structure (*Paninski et al., 2007*; *Fairhall and Sompolinsky, 2014*). Our main finding was that responses of PWNs, recorded as an awake mouse actively explores an object with its whiskers, can be predicted from the forces acting on the whiskers. Given that, for each unit, we were attempting to predict the entire ~70 s time course of activity, the variability of the behaviour of untrained mice (*O'Connor et al., 2010a*), and the lack of trial-averaging as a noise reduction strategy, it is remarkable that we found model prediction correlation coefficients up to 0.88. A challenge of studying neural coding under unconstrained, awake conditions is that sensory variables tend to correlate. A valuable feature of the GLM training procedure is that it takes such correlations into account. We found that, although whisker angle predicted spikes for a subset of units, this effect was very largely explained by a curvature-coding model, together with the correlation between angle and curvature.

### Mechanical framework for tactile coding

Pushing a whisker against an object triggers spiking in many PWNs (*Szwed et al., 2003*; *Szwed et al., 2006*; *Leiser and Moxon, 2007*). Biomechanical modelling by Hartmann and co-workers accounts for this by a framework where the whisker is idealised as an elastic beam, cantilever-mounted in the skin (*Birdwell et al., 2007*; *Quist et al., 2014*). When such a beam pushes against an object, the beam bends, causing reaction forces at its base. Our data are in striking agreement with the general suggestion that mechanoreceptor activity is closely related to such reaction forces. Our results show that curvature change associated with contact-induced whisker bending, and acceleration associated with whisker rotation, predict PWN spiking. Our results also provide a mechanical basis for previous findings: our finding of subtypes of curvature-only and curvature-acceleration PWNs is consistent with previous reports of 'touch' and 'whisking-touch' units (*Szwed et al., 2003*; *2006*). Thus, a common framework accounts for diverse PWN properties.

Our finding that whisker angle predicts PWN spikes poorly indicates that whisker angle can change without modulating mechanotransduction in the follicle. This is consistent with evidence that, during artificial whisking, the follicle-shaft complex moves as a rigid unit (*Bagdasarian et al., 2013*). In apparent contrast, previous studies using passive stimulation in anaesthetised animals have consistently reported a tight relationship between whisker kinematics and PWN response. In the cantilever whisker model, passively induced changes in whisker angle correlate highly with whisker bending. We confirmed that this applies to real whiskers in vivo and demonstrate that moment-sensitive units can thereby appear angle-tuned. In this way, moment-encoding can account for primary neuron responses not only during active touch but also under passive stimulation. More generally, our results highlight the importance of studying neurons under natural, active sensing conditions.

In this study, we considered PWN encoding under conditions of pole contact, since this is well-suited to reaction force estimation (*O'Connor et al., 2010a*; *Pammer et al., 2013*) and involves object-stimulus interactions on a ~100 ms time-scale that is conducive to single-trial analysis. Since whisker bending is ubiquitous in whisking behaviour, it is likely that our finding of curvature sensitivity is a general one. However, prediction performance varied across units, suggesting that other force components may also be encoded. Other experimental conditions – for example, textured surfaces – may involve multiple force components (*Quist and Hartmann, 2012*; *Pammer et al., 2013*;

*Bagdasarian et al., 2013*) and/or encoding of information by spike timing on a finer time-scale (*Panzeri et al., 2001*; *Petersen et al., 2001*; *Arabzadeh et al., 2005*; *Bale et al., 2015*).

It is axiomatic that mechanoreceptors are sensors of internal forces acting in the tissue within which they are embedded (*Abraira and Ginty, 2013*) and therefore valuable to be able to measure mechanical forces in the awake, behaving animal. In general, including the important case of primate hand-use, the complex biomechanics of skin makes force-estimation difficult (*Phillips and Johnson, 1981*). In contrast, for whiskers, the quasi-static relationship is relatively simple: the bending moment on a whisker is proportional to its curvature. This has the important implication that reaction forces can be directly estimated from videography in vivo (*Birdwell et al., 2007*; *O'Connor et al., 2010a*; *Pammer et al., 2013*). Our results are the first direct demonstration that such reaction forces drive primary sensory neuron responses – likely involving Piezo2 ion channels (*Woo et al., 2014*; *Poole et al., 2015*; *Whiteley et al., 2015*) – and provide insight into how sensitivity to touch and self-motion arises in the somatosensory pathway (*Szwed et al., 2003*; *Yu et al., 2006*; *Curtis and Kleinfeld, 2009*; *Khatri et al., 2009*; *O'Connor et al., 2010b*; *Huber et al., 2012*; *Petreanu et al., 2012*; *Peron et al., 2015*).

## Moment-based computations in tactile behaviour

Extraction of bending moment is a useful first step for many tactile computations. Large transients in bending moment signal object-touch events, and the magnitude of bending is inversely proportional to the radial distance of contact along the whisker (*Solomon and Hartmann, 2006*). As illustrated by our results on the statistics of active touch, if integrated with cues for whisker self-motion, whisker bending can be a cue to the 3D location of an object (*Szwed et al., 2003*; *2006*; *Birdwell et al., 2007*; *Bagdasarian et al., 2013*; *Pammer et al., 2013*). Bending moment can permit wall following (*Sofroniew et al., 2014*) and, if integrated across whiskers, can in principle be used both to infer object shape (*Solomon and Hartmann, 2006*) and to map the spatial structure of the environment (*Fox et al., 2012*; *Pearson et al., 2013*).

## Summary and conclusion

We have shown that the responses of primary whisker neurons can be predicted, during natural behaviour that includes active motor control of the sense organ, from forces acting on the whiskers. These results provide a bridge linking receptor mechanisms to behaviour.

# Materials and methods

All experimental protocols were approved by both United Kingdom Home Office national authorities and institutional ethical review.

## Surgical procedure

Mice (C57; N=10; 6 weeks at time of implant) were anesthetized with isoflurane (2% by volume in $O_2$), mounted in a stereotaxic apparatus (Narishige, London, UK) and body temperature maintained at 37°C using a homeothermic heating system. The skull was exposed and a titanium head-bar (19.1 × 3.2 × 1.3 mm; *O'Connor et al., 2010a*) was first attached to the skull ~1 mm posterior to lambda (Vetbond, St. Paul, MN), and then fixed in place with dental acrylic (Lang Dental, Wheeling, IL). A craniotomy was made (+0.5 to -1.5 mm posterior to bregma, 0-3 mm lateral) and sealed with silicone elastomer. Buprenorphine (0.1 mg/kg) was injected subcutaneously for postoperative analgesia and the mouse left to recover for at least 5 days.

## Behavioural apparatus

Mice were studied in a pole exploration apparatus adapted from *O'Connor et al., 2010a* , but were not trained on any task. A mouse was placed inside a perspex tube (inner diameter 32 mm), from which its head emerged at one end, and immobilised by fixing the head-bar to a custom mount holder. The whiskers were free of the tube at all times. The stimulus object was a 1.59 mm diameter metal pole, located ~3.5 mm lateral to the mouse's snout. To allow control of its anterior/posterior location, the pole was mounted on a frictionless linear slide (NDN 2-50.40, Schneeberger, Roggwil, Germany) and coupled to a linear stepper motor (NA08B30, Zaber, Vancouver, Canada). To allow

vertical movement of the pole into and out of range of the whiskers, the pole/actuator assembly was mounted on a pneumatic linear slide (SLS-10-30-P-A, Festo, Northampton, UK), powered by compressed air. The airflow was controlled by a relay (Weidmüller, Richmond, VA). In this way, the pole moved rapidly (~0.15 s) into and out of range of the whiskers. The apparatus was controlled from Matlab via a real-time processor (RX8, TDT, Alachua, FL).

## Electrophysiology

We recorded the activity of PWNs from awake mice in the following way. To permit reliable whisker tracking (see below), before each recording session, A, B and E whisker rows were trimmed to the level of the fur, under brief isoflurane anaesthesia. The trigeminal ganglion was targeted as previously described (*Bale et al., 2015*). The silicone seal was removed and a 3/4 shank tungsten microelectrode array (FHC, Bowdoin, ME, recording electrodes 8 MΩ at 1 kHz, reference 1 MΩ; tip spacing ~500 µm) was lowered through the brain (angle 4° to vertical in the coronal plane) using a micromanipulator (PatchStar, Scientifica, Uckfield, UK) under isoflurane anaesthesia. Extracellular potentials were pre-amplified, digitised (24.4 kHz), filtered (band pass 300–3000 Hz) and acquired continuously to hard disk (RZ5, TDT). The trigeminal ganglion was encountered 6–7 mm vertically below the pial surface and whisker-response units identified by manual deflection of the whiskers with a small probe. Once a well-isolated unit was found, the whisker that it innervated (the 'principal whisker', PW) was identified by manual stimulation. To define the PW, we deflected not only untrimmed whiskers, but also the stubs of the trimmed whiskers. Any unit whose PW was a trimmed whisker was ignored. At this point, anaesthesia was discontinued. Once the mouse was awake, we recorded neuronal activity during repeated presentations of the pole ('trials'). Before the start of each trial, the pole was in the down position, out of reach of the whiskers. The pole was first moved anterior-posteriorly to a position chosen randomly out of a set of 11 possible positions, spanning a range ± 6 mm with respect to the resting position of the base of the PW. A trial was initiated by activating the pneumatic slide relay, thus moving the pole up into the whisker field, where it remained for 3 s before being lowered. At the end of a recording session, the microelectrode array was withdrawn, the craniotomy sealed with silicone elastomer, and the mouse returned to its home cage.

## High-speed videography

Using the method of *O'Connor et al. (2010a)* to image whisker movement/shape, whiskers ipsilateral to the recorded ganglion were illuminated from below using a high-power infrared LED array (940 nm; LED 940-66-60, Roithner, Vienna, Austria) via a diffuser and condensing lens. The whiskers were imaged through a telecentric lens (55-349, Edmunds Optics, Barrington, NJ) mounted on a high speed camera (LTR2, Mikrotron, Unterschleissheim, Germany; 1000 frames/s, 0.4 ms exposure time). The field of view of the whiskers was 350×350 pixels, with pixel width 0.057 mm.

## Response to touch and non-touch events

Mouse whisking behaviour during the awake recording was segmented into 'touch', and 'non-touch' episodes. Touches between the PW of each unit and the pole were detected manually in each frame of the high-speed video. A frame was scored as touch if no background pixels were visible between the pole silhouette and the whisker. Any frame not scored as a touch was scored as non-touch. Touch and non-touch firing rates for a given unit were computed by averaging activity over all corresponding episodes.

## Whisker tracking

Since the trigeminal ganglion lacks topography, it is difficult to target units that innervate a specific whisker, and therefore desirable for a whisker tracker to be robust to the presence of multiple rows of whiskers. However, since neurons in the ganglion innervate individual whiskers, it is sufficient to track only one whisker (the PW) for each recorded neuron. To extract kinematic/mechanical whisker information, we therefore developed a whisker tracker ('WhiskerMan'; *Bale et al., 2015*) whose design criteria, different to those of other trackers (*Perkon et al., 2011*; *Clack et al., 2012*), were to: (1) be robust to whisker cross-over events; (2) track a single, target whisker; (3) track the proximal segment of the whisker shaft. The shape of the target whisker segment was described by a quadratic

Bezier curve **r**(*t,s*) (a good approximation away from the zone of whisker-object contact; *Quist and Hartmann, 2012*; *Pammer et al., 2013*): **r**(*t,s*) = [*x*(*t,s*), *y*(*t,s*)], where *x*, *y* are horizontal/vertical coordinates of the image, *s* = [0,..,1] parameterises (*x,y*) location along the curve and *t* is time. We fitted such a Bezier curve to the target whisker in each image frame using a local, gradient-based search. The initial conditions for the search were determined by extrapolating the solution curves from the previous two frames, assuming locally constant, angular velocity. The combination of the low-parameter whisker description and the targeted, local search made the algorithm robust to whisker cross-over events. The 'base' of the target whisker was defined as the intersection between the extrapolated Bezier curve and the snout contour (estimated as described in *Bale et al., 2015*). The solution curve in each frame was visually checked and the curves manually adjusted to correct occasional errors.

## Estimation of kinematic/force parameters

The whisker angle ($\theta$) in each frame was measured as the angle between the tangent to the whisker curve at the base and the anterior-posterior axis (*Figure 1C*). Whisker curvature ($\kappa$) was measured at the base as $\kappa = \frac{x'y'' - x''y'}{(x'^2 + y'^2)^{3/2}}$, where *x'*, *y'* and *x''*, *y''* are the first and second derivatives of the functions *x(s)* and *y(s)* with respect to *s* (*Figure 1C*). Since reaction force at the whisker base reflects changes in whisker curvature, rather than the intrinsic (unforced) curvature (*Birdwell et al., 2007*), we computed 'curvature change' $\Delta\kappa = \kappa - \kappa_{int}$, where $\kappa_{int}$, the intrinsic curvature, was estimated as the average of $\kappa$ in the first 100 ms of the trial (before pole contact; *O'Connor et al., 2010a*). During free whisking, whisker angle oscillated with the characteristic whisking rhythm, but curvature changed little. The small changes in whisker curvature during free whisking were consistent with torsional effects (*Knutsen et al., 2008*). We estimated the number of whisking cycles from the duration of touch/non-touch episodes and the whisking frequency: median 419 whisking cycles per unit during touch periods; 415 during non-touch periods.

Under conditions of whisking against a smooth surface, such as in the present study, the quasistatic framework of *Birdwell et al. (2007)* applies. $\Delta\kappa$, measured, at the base of a whisker, in the horizontal plane, is proportional to the component of bending moment in that plane. We used $\Delta\kappa$ as a proxy for bending moment. Bending moment (*M*), Axial ($\vec{F}_{ax}$) and lateral force ($\vec{F}_{lat}$) at the whisker base were calculated, during periods of whisker-pole contact, using the method of *Pammer et al. (2013)*, using published data on areal moment of inertia of mouse whiskers (*Pammer et al., 2013*), along with whisker-pole contact location (see *Figure 1—figure supplement 2* for details). Pole location, in the horizontal plane, in each frame, was identified as the peak of a 2D convolution between the video image and a circular pole template. To localise whisker-pole contact, the whisker tracker was used to fit the distal segment of the whisker close to the pole, seeded by extrapolation from the whisker tracking solution for the proximal whisker segment, described above. Whisker-pole contact location was defined as the point where this distal curve segment was closest to the detected pole centre. Pole and contact locations were verified by visual inspection.

As expressed by Newton's second law of rotational motion, the moment – or torque – of a rigid body, rotating in a plane, is proportional to the body's angular acceleration. During free whisking, a whisker behaves approximately as a rigid body and, for the whiskers considered in this study, their motion is predominantly in the horizontal plane (*Bermejo et al., 2002*; *Knutsen et al., 2008*). Thus, to assess whether such a moment is encoded by PWNs, we measured angular whisker acceleration during free whisking as a proxy. Acceleration was calculated from the whisker angle time series after smoothing with a Savitzky-Golay filter (polynomial order 5; frame size 31 ms).

Push angle – the change in angle as a whisker pushes against an object - was measured during touch epochs. For each touch episode, we determined the value of the angle in the frame before touch onset and subtracted this from the whisker angles during the touch.

## Passive whisker deflection

To determine how whiskers move/bend in response to passive deflection under anaesthesia, a mouse was anesthetized (isoflurane 2%) and placed in the head-fixation apparatus. Individual whiskers (C2 and C5 trimmed to 5 mm) were mechanically deflected using a piezoelectric actuator as previously described (*Bale et al., 2013*; *2015*). All other whiskers were trimmed to the level of

the fur. Each whisker, in turn, was inserted into a snugly fitting plastic tube attached to the actuator, such that the whisker entered the tube 2 mm from the face. Two stimuli were generated via a real-time processor (TDT, RX8): (1) a 10 Hz trapezoidal wave (duration 3 s, amplitude 8°); (2) Gaussian white noise (duration 3 s, smoothed by convolution with a decaying exponential: time constant 10 ms; amplitude SD 2.1°). During the stimulation, the whiskers were imaged as detailed above (1000 frames/s, 0.2 ms exposure time).

## Electrophysiological data analysis

### Spike sorting

Single units (N=20) were isolated from the extracellular recordings as previously described, by thresholding and clustering in the space of 3–5 principal components using a mixture model (*Bale and Petersen, 2009*). A putative unit was only accepted if (1) its inter-spike interval histogram exhibited a clear absolute refractory period and (2) its waveform shape was consistent between the anaesthetised and awake phases of the recording.

### Responses to whisking without touch

To test whether a unit responded to whisking itself, we extracted non-touch episodes as detailed above and computed time series of whisking amplitude and phase by band-pass filtering the whisker angle time series (6–30 Hz) and computing the Hilbert transform (*Kleinfeld and Deschênes, 2011*). Amplitudes were discretised (30 equi-populated bins) and the spiking data used to compute amplitude tuning functions. Phases for bins where the amplitude exceeded a given threshold were discretised (8 equi-populated bins) and used to construct phase tuning functions. To determine whether a unit was significantly amplitude-tuned, we fitted a regression line to its amplitude tuning curve and tested whether the slope was statistically significantly different to 0 (p=0.0025, Bonferroni-corrected). To determine whether a unit was significantly phase-tuned, we computed the maximum value of its phase tuning curve and compared this to the distribution of maxima of chance tuning functions. Chance tuning functions were obtained by randomly shifting the recorded spike sequences by 3000–8000 ms and recomputing tuning functions (500 times). A unit was considered phase-tuned if its tuning function maximum (computed using amplitude threshold of 2°) exceeded the 95th percentile of the shuffled distribution.

Acceleration tuning curves were quantified, for each unit, as follows. First, an acceleration tuning curve was estimated (as above). Units typically responded to both positive and negative accelerations, but with unequal weighting between them. To capture this, we fitted the following regression model to the tuning curve:

$$r_i = \mu_0 + \mu_1 a_i + \mu_2 \Delta_i a_i$$

Here, for each bin $i$ of the tuning curve, $r_i$ was the firing rate and $a_i$ was the acceleration; $\mu_0$, $\mu_1$ and $\mu_2$ were regression coefficients; the term $\Delta_i$ ($\Delta_i=1$ if $a_i<0$, $\Delta_i=0$ otherwise) allowed for asymmetric responses to negative and positive acceleration. Based on its best-fitting regression coefficients (p=0.05), units were classified as: having 'preference for negative acceleration', if $\mu_2$ was significantly >0; having 'preference for positive acceleration', if $\mu_2$ was significantly <0; as having 'no preferred direction' if both $\mu_1$ was significantly >0, and $\mu_2$ was not significantly different from 0; and as 'not acceleration sensitive' if neither $\mu_1$ nor $\mu_2$ were significantly different from 0.

### Generalised Linear Model (GLM)

To investigate how well PWNs encode a given sensory variable (e.g., whisker angle, curvature), we fitted single unit activity to a GLM (*Nelder and Wedderburn, 1972*; *Truccolo et al., 2005*; *Paninski et al., 2007*), using methods similar to *Bale et al., 2013*. For each unit, a 'stimulus' time series ($x$) (whisker angle or whisker curvature change) and a simultaneously recorded spike time series ($n$) were discretized into 1 ms bins: $x_t$ and $n_t$ denote respectively the stimulus value and spike count (0 or 1) in bin $t$.

GLMs express how the expected spike count of a unit depends both on the recent stimulus history and on the unit's recent spiking history. The standard functional form of the model we used was:

$$y_t = f\left(\vec{k}^{\rightarrow T} \vec{x_t} + \vec{h}^{\rightarrow T} \vec{n_t} + b\right) \tag{1}$$

Here $n_t$, the output in bin $t$, was a Bernoulli (spike or no-spike) random variable. The probability of a spike in bin $t$, $y_t$, depended on three terms: (1) the dot product between the stimulus history vector $\vec{x}_t = (x_{t-Lk+1}, \ldots, x_t)$ and a 'stimulus filter' $\vec{k}$ (length $L_k$ = 5); (2) the dot product between the spike history vector $\vec{n} = (n_{t-Lh+1}, \ldots, n_t)$ and a 'spike history filter' $\vec{h}_t$ (length $L_h$= 2); (3) a constant bias $b$, which sets the spontaneous firing rate. $f(\bullet)$ was the logistic function $f(z) = (1 + e^{-z})^{-1}$. The preferred direction of the GLM is determined by the sign of the stimulus filter. Positive (negative) $k$ coefficients tend to make positive (negative) stimuli trigger spikes. Since we found that GLM performance was just as good with $L_k$ = 1 as $L_k$ = 5 (*Figure 2—figure supplement 1C*), we used results from the $L_k$ = 1 case to define selectivity to curvature change direction: positive $k$ implies selectivity for positive curvature change; negative $k$ selectivity for negative curvature change. When a whisker pushed against an object during protraction, curvature increased; when it pushed against an object during retraction, it decreased.

To consider whether units might encode multiple sensory variables (e.g., both whisker angle and whisker curvature change), we used a GLM with multiple stimulus history terms, one for each sensory variable:

$$y_t = f\left(\vec{k_1}^{\rightarrow T} \vec{x_{t;1}} + \vec{k_2}^{\rightarrow T} \vec{x_{t;2}} + \vec{h_t}^{\rightarrow T} \vec{n*} + b\right)$$

Here the indices 1, 2 label the sensory variables.

Training and testing of the GLM were done using a cross-validation procedure. For each unit, half of the trials were assigned randomly to a training set and half to a testing set. The training set was used to fit the parameters ($\vec{k}$, $\vec{h}$ and $b$), while the testing set was used to quantify the similarity between the spike train of the recorded unit and that predicted by the GLM. GLM fitting was achieved by finding the parameter values ($\vec{k}$, $\vec{h}$ and $b$), which minimized a cost function consisting of the sum of the negative log-likelihood and a regularizing term $-\alpha\|\vec{k}\|^2$. For all units, model prediction performance on the test set was robust to variation of $\alpha$ over several orders of magnitude: $\alpha$ was therefore set to a standard value of 0.01. To quantify the performance of the model, the sensory time series of the testing set was used as input to the best-fitting GLM to generate a 'predicted' spike train in response. Both real and predicted spike trains were then smoothed by convolution with a 100 ms box-car filter and the similarity between them quantified by the Pearson correlation coefficient (PCC). For each unit, the entire training/testing procedure was repeated for 10 random choices of training/testing set and the final prediction accuracy defined as the median of the 10 resulting PCC values. Data from these 10 samples were also used to test whether an individual unit exhibited statistically significant prediction performance for different sensory features. To test whether the results were robust to the smoothing time-scale, the above procedure was repeated for a range of box-car smoothing filters (1, 5, 10, 20, 50, 70 ms). To test whether a given 'actual' PCC was statistically significant, we tested the null hypothesis that it could be explained by random firing at the same time-averaged rate as that of the recorded unit. To this end, the recorded spike sequences were randomly shifted by 3000–8000 ms and the training/testing procedure above applied to this surrogate data. This was repeated 10 times and the resulting chance PCCs compared to the actual PCC using a signed-rank test, p=0.0025 (Bonferroni-corrected). This analysis was used to classify units as being 'curvature-sensitive'.

## Quadratic GLM

To test whether the units might exhibit nonlinear dependence on the stimulus parameters, we adapted the GLM defined above (*Equation 1*) to include quadratic stimulus variables (*Rajan et al., 2013*). This was important to assess whisker angular acceleration during free whisking, since a subset of units exhibited U-shaped acceleration tuning functions (*Figure 3B*). Given a stimulus time series $x_t$, the quadratic stimulus history vector was $[x_{t-Lk+1}, \ldots, x_t, x^2_{t-Lk+1}, \ldots, x^2_t]$. Fitting methods were otherwise identical to those detailed above.

## Effect of angle-curvature correlations on apparent neuronal stimulus encoding in the passive stimulation protocol

If, in a given recording, sensory variable X correlates with sensory variable Y, a neuron responsive purely to X will tend to appear tuned to Y. To investigate whether such an effect might produce apparent sensitivity to whisker angle in the passive stimulation paradigm, we simulated the response of curvature-tuned neurons to the whisker curvature change time series measured during passive white noise stimulation. To minimise free parameters, constrained GLMs (4 free parameters) were used, sensitive either to instantaneous curvature ($\vec{k} = [\gamma]$) or to its first order derivative ($\vec{k} = \gamma[-1\ 1]$), where $\gamma$ was a signed, gain parameter. Parameters ($\vec{h}$, $b$, $\gamma$) were adjusted to produce two spike trains (one for training, the other for testing) with a realistic white noise induced firing rate (~50 spikes/s; *Bale et al., 2013*). We then attempted to predict the simulated, curvature-evoked (training) spike train by fitting GLMs (length 5 stimulus filter, 8 free parameters) using as input either angle or curvature change. Cross-validated model accuracy was computed as the PCC between the predicted spike train and the testing spike train (both smoothed by convolution with a 5 ms box-car).

## Effect of single-trial approach on GLM prediction performance

The objective of encoding models, such as GLMs, is to obtain an accurate description of the mapping between a stimulus and the neuronal spike trains it evokes. Since the random component of a neuron's response is inherently unpredictable, the best any model can do is to predict the probability of the spike train. To enable this, encoding models have generally (with few exceptions; *Park et al., 2014*) been applied to a 'repeated-trials' paradigm, where a stimulus sequence (e.g., frozen white noise) is repeated on multiple 'trials' (*Arabzadeh et al., 2005*; *Lottem and Azouz, 2011*; *Bale et al., 2013*; *Petersen et al., 2008*; *Pillow et al., 2008*). Model accuracy can then be quantified, largely free of contamination from random response variability, by comparing (using PCC or otherwise) the trial-averaged response of the model to the trial-averaged response of the neuron.

In contrast, in the present study of awake, actively whisking mice, the precise stimulus (time series of whisker angle/curvature) was inevitably different on every pole presentation: there were no precisely repeated trials to average over. Our standard model performance metric (PCC) was computed by comparing the response on a single long, concatenated 'trial' with the corresponding GLM predicted response. Such a PCC is downwards biased by random response variability.

To gauge the approximate magnitude of this downward bias, we used a simulation approach. By simulating the response of model neurons, we could deliver identical, repeated trials and thereby compare model prediction performance by a metric based on trial-averaging with that based on the single-trial approach. To this end, for each recorded unit, we used the best-fitting curvature change GLM to generate 100 trials of spike trains evoked by the curvature time series measured for that unit. Data from the first of these trials was used to fit the parameters of a minimal 'refitted GLM' (stimulus filter length 1, spike history filter length 2; bias; total 4 free parameters), and the single-trial performance quantified, using the approach of the main text (*Figure 2—figure supplement 1B*, left). Next, we used the refitted GLM to generate 100 repeated trials of spike trains evoked by the curvature time series. Repeated-trials performance was then quantified as the PCC between PSTHs obtained by trial-averaging (*Figure 2—figure supplement 1B*, right).

## Acknowledgements

We thank S Fox, M Humphries, M Loft, R Lucas, M Montemurro and M Maravall for comments on the manuscript/discussion; K Svoboda for sharing behavioural methods; G Caspani, K Chlebikova, B Nathanson and R Twaites for assistance with whisker tracking.

## Additional information

### Funding

| Funder | Grant reference number | Author |
|---|---|---|
| Biotechnology and Biological | BB/L007282/1 | Rasmus Strange Petersen |

Sciences Research Council

| Wellcome Trust | 097820/Z/11/B | Rasmus Strange Petersen |
| Medical Research Council | MR/L01064X7/1 | Rasmus Strange Petersen |

The funders had no role in study design, data collection and interpretation, or the decision to submit the work for publication.

## Author contributions

DC, Designed the study, Performed the experiments, Analyzed the data, Developed the experimental methods, Wrote the manuscript.; MHE, Analyzed the data, Developed the experimental methods; MRB, Developed the experimental methods.; AE, Performed the experiments, Developed the experimental methods.; RSP, Designed the study, Analyzed the data, Developed the experimental methods, Wrote the manuscript.

## Author ORCIDs

Dario Campagner, http://orcid.org/0000-0001-9016-4575
Andrew Erskine, http://orcid.org/0000-0003-4392-1873

## Ethics

Animal experimentation: All experimental protocols were approved by both United Kingdom Home Office national authorities and institutional ethical review. Project licence: 40/3420.

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
