## [Decision Letter]

Thank you for submitting your work entitled "Prediction of primary somatosensory neuron activity during active tactile exploration" for peer review at *eLife*. Your submission has been favorably evaluated by Eve Marder (Senior editor), a Reviewing editor, and three reviewers.

The Reviewing editor and the reviewers agree that this is a potentially very important work that defines the features of whisking and vibrissa contact that cause a trigeminal ganglion neuron to spike. Further, while this latter topic has been explored extensively in anesthetized animals, there is limited work in the awake case. Thus a study of parameters that lead to spiking in the awake case would be a welcome contribution to the vibrissa field. Yet there are major weaknesses in the current submission that *must* be addressed, with additional analysis and potentially new experiments, before we can proceed further.

Major issues:

The choice of the angular variable was absolute angle – which is likely to be a poor choice. The analysis should be redone in terms of deflection angle. The change in position of the mystacial pad also needs to be taken into account in the analysis, as noted by multiple reviewers. Related to this is a need to re-evaluate claims, as noted by reviewer 3 "[…]only 50% of the data shows a drop in performance when angle is used instead of curvature. The other 50% seems to perform as well for angle as it does for curvature. The conclusion that curvature is therefore the single most important parameter seems not to be supported by the data." This requires reanalysis, yet also may require some additional data to increase the spike count.

The application of the GLM was done with relatively few spikes, about 550 or 275 each for training and testing sets for 8 feature parameters and an unstated number of spike history parameters, which gives the editor pause. It would be thus imperative for the author to show the feature vector and the history term for at least a few units, as opposed to just predictions. In fact, the feature vector is a main point of such an analysis. Further, while the recorded data show clear cycle-by-cycle whisks (Figure 1), and "whisking" cells in the trigeminal ganglion faithfully respond on a cycle-by-cycle basis, these fast changes do not appear in the predicted spike rates of Figure 2. The reason for this omission, as pointed out by Reviewer #1, needs to be explained.

Further on the topic of analysis, Reviewer #2 notes "The authors use the poor performance of angle GLMs during active pole exploration as evidence that curvature changes are what drive PWNs. But it is known that touch dominates PWN spiking responses, so including touch periods when assessing angle GLM decoding will trivially result in very low GLM angle performance. More interesting would be comparing the performance of angle GLMs during non-touch, free whisking periods with performance of curvature GLMs during touch periods." The critical issue is to determine if there is a big difference between passive and active states or if, within strong statistics, there is not a big difference. This requires reanalysis, yet also may require some additional data to increase the spike count.

Please read through the attached thorough and very thoughtful reviews by three of your colleagues and please address all of the issues raised in a cover letter to accompany the resubmitted manuscript.

*Reviewer #1:*

This is an elegant work, addressing a crucial question – what do sensory neurons code during active exploration and touch – in a professional manner. The paradigm is simple and clear, and the paper is well written (for the most part), expressing clear thinking and straight-forward reasoning. This impressive work can potentially advance the understanding of sensory coding significantly. Yet, in its current form there is a danger that the paper will instead increase the confusion in the field – this is due to several major flaws that need to be carefully addressed.

1) The choice of the angular variable for analysis. The authors analyze the angle of the whisker relative to the head – let's call this the "absolute angle" here. There are 2 problems with it. One, portions of the pad rotate with significant angles during active whisking such that the absolute angle of a whisker changes but this has no effect on the shaft-follicle mechanical interactions (the entire complex moves together). This can be seen in the supplementary video of the paper, when examining the pad. Thus, the angle should be measured relative to the pad surrounding the whisker and not to the fixed head. Second, the relevant angular variable in the sensory coding game is most likely the *change* in angle upon contact. Both the "push angle" (Quist and Hartman, 2012; Bagdasarian et al., 2013) and the "Angle absorption" (Bagdasarian et al., 2013) carry meaningful information. This analysis of relative angular changes upon touch will also make the angle-curvature comparison more symmetric (currently the change in curvature upon touch is compared with the absolute angle).

2) After reading through the Results section it turns out that this study actually (re) revealed two types of cells – those termed by Szwed et al. (2003) as "Whisking" cells and "Touch" cells. As in Szwed et al. (2003), the former respond to whisking in air and are sensitive to the phase of whisking and the latter respond to touch and are sensitive to curvature changes. This fact should be described at the outset (Abstract, Introduction, Discussion) and compared with the relevant previous reports.

What seems to be missing here are two complementary analyses – the sensitivity of "Whisking" cells to touch and of "Touch" cells to whisking in air. Thus, the fractions of pure whisking and touch cells, and that of a combined "whisking-Touch" type (Szwed, 2003) is not clear. True, the cell count is not high (I believe it is total of 20, although this was hard to dig – please state it at the outset) but the cost of this should not be in flattening all types to one common denominator.

Importantly, the point in the paper where the reader realizes that half of the cells are "Whisking" cells is a confusing point, reflecting back on the initial analysis. For clarity, the separation between cell types should be clarified at the beginning.

3) The Abstract statement "[…]we found that primary neuron responses were poorly predicted by kinematics but well-predicted by rotational forces acting on the whisker[…]" is not supported by the data. In fact, the insisting on a single mechanical variable does not make much sense, is not convincing and, as said, is not consistent with the data presented in the paper. I strongly recommend re-considering it. First, the paper shows that half of the cells (the whisking cells) are actually sensitive to a kinematic variable, acceleration. Indeed, it is associated with force but aren't all kinematic changes associated with forces? Also, selecting whisker acceleration instead of other angular variables, such as phase and velocity, and even angle itself, seems to be arbitrary. As for the Touch cells, indeed the curvature is correlated with various angular variables, but the parameters of these correlations depend on the interactions with external objects (see Bagdasarian et al., 2013), interactions that are not investigated here. In fact, Bagdasarian et al. showed that relying on a single mechanical variable must lead to ambiguity about external features.

4) The paper deals only with slow dynamics of coding – in time scales of seconds and resolution > 100 ms. Analysis at higher temporal resolutions (as was impressively done by the Petersen lab previously) is probably not possible in the current challenging setup of TG recording in awake animals. Yet, perceptual processing depends crucially on within-cycle millisecond time scales. This should be emphasized at the outset and discussed in relation to candidate sensory variables and relevant external features. It seems that while this slow time course may be relevant to features such as object radial distance (in the case of touch) and intensity of whisking (combination of whisking amplitude and frequency, which determine average acceleration throughout the cycle – see Figure 3), but not object azimuthal position, texture or shape and not phase within the whisking cycle. Also, the choice of 100 ms should be justified, and the dependency of the results on this choice should be described.

5) Figure 1 show a whisker that pushes against the object during retraction. The video and Figure 1 shows the "standard" contact, during protraction. The authors should make it clear whether their analysis was based on both directions. If so, this comment becomes a major one – the authors must include the direction as one of the analyzed variables and describe the dependency of the various findings on it. Also, curvatures are very strong in this study (Figure 2, movie). Please refer to it and compare to free-head conditions in which often the minimal impingement principle (Prescott et al., 2013) applies. Please discuss the implications on the predominance of curvature coding in this study.

*Reviewer #2:*

Key findings:

1) PWNs are relatively insensitive to absolute whisker angle but highly sensitive to curvature change.

2) The degree to which PWNs are tuned to curvature change predicts their response to inertial force during free whisking.

These results are well supported by the data, and the data is valuable, nicely collected and presented. However, the results don't change the general understanding of PWN coding and thus are not novel. The paper focuses on overturning a straw-man characterization of the literature, that PWNs are tuned to absolute whisker angle, not deflection forces.

It is unfair to characterize the current results as "at odds with passive stimulation studies (Gibson 1983, Lichtenstein 1990[…])". The classic studies refer to PWN tuning to angle of deflection not absolute angle. These particular studies had no ability to assess PWN tuning in the absence of deflection. In Bale (2013), again the positional tuning was in the context of positional deflection not free whisking angle. Indeed, Leiser (2007) showed that firing rates are 10x higher in PWNs during contact than during awake free whisking. The logical interpretation of this and many other cited studies of PWN coding is that deflection-induced forces (often quantified as deflection angle) are the primary driver of PWN spiking, not whisker position absent deflection.

The authors use the poor performance of angle GLMs during active pole exploration as evidence that curvature changes are what drive PWNs. But it is known that touch dominates PWN spiking responses, so including touch periods when assessing angle GLM decoding will trivially result in very low GLM angle performance. More interesting would be comparing the performance of angle GLMs during non-touch, free whisking periods with performance of curvature GLMs during touch periods.

In the study, active touch occurs at multiple pole positions, while passive deflections have only one starting position. Thus the comparison of curvature and angle coupling between active and passive conditions (Figure 4) is apples to oranges. For example, if the mouse must position his whisker 10 degrees more protracted to contact the pole in one position vs. another during active sensing, the correlation between angle and curvature will be degraded when averaged across pole positions. Including non-touch periods in the analysis further degrades the correlation. Thus the poor cross-correlation for the awake condition in Figure 4 is trivial.

The more interesting and fair comparison is the extent to which active control of whisker position impacts the relationship between curvature and push angle. Push angle is defined as the angle through which the whisker is rotated into the object (see Quist and Hartmann, 2012 or Hires, 2013 for details). Active control could alter the rigidity of the follicle, impacting follicle stresses and thus spiking activity of PWNs. This should be detectable via comparing the difference in push angle/curvature coupling (i.e. the slope of touch trajectories in 4E, assuming curvature was measured at the same radial distance) between active and passive states.

Additional comments:

The data in the paper are interesting and do have potential to address some open questions that would increase the importance and novelty of the work. Some possible ideas that reanalysis could address, (in order of increasing interest):

1) Do PWNs that are tuned to acceleration direction show the same directional selectivity to deflection direction?

2) Do force components (Faxial, bending moment) differentially drive PWNs?

3) Do PWN responses to passive vs. active touch exhibit different sensitivity to deflection angle or whisker curvature change?

Detailed justification:

1) This would be a simple expansion of the analysis of Figure 3 to show correlation of directional tuning between touch and whisking across the population of whisking sensitive neurons. This would make the Figure 3 result more compelling.

2) Using Faxial and Bending Moment as independent predictors in a GLM could determine if PWNs specialize for these components during active touch. Longitudinal deflection of whiskers causes robust responses in PWNs (Zucker and Welker 1969, Stuttgen 2008). Axial and lateral/moment ratios are used for radial object localization (Solomon 2011, Pammer 2013). This could bridge those physiology and behavior results.

3) Quantifying a non-trivial difference between passive and active touch, particularly if reflected in spiking activity would make the paper much more interesting. In cortex differences have been seen (e.g. airpuff of whiskers when awake elicits much weaker dendritic responses in Figure 1—figure supplement 1 than active touch, despite the much greater deflections air puffs evoke Xu 2012 Nature). Are these differences inherited from PWNs due to different mechanical coupling or sensitivity between these states?

*Reviewer #3:*

The manuscript of Campagner et al. investigates the whisker parameters (angle and curvature) that allow reliable prediction of spiking of primary whisker neurons upon passive or active touch. The manuscript is potentially interesting, although I have some concern about experimental setup and the validity of comparisons between passive and active conditions. Additionally, even though curvature reliably predicts spiking in awake rats for a subset of the data, the range of quantified reliability is large and not discussed.

1) The major conclusion (curvature much better predicts spiking than angle) is based predominantly on Figure 2. The full range of reliability measures for curvature is 0.1 – 0.9. The authors put a lot of emphasis on the fraction of high values (max 0.88), but completely ignore the lower measures. Vice versa, the high values for angle GLMs are only briefly mentioned and emphasis put on poor predicting values. It seems very relevant to discuss the entire range for both conditions. Additionally, only 50% of the data shows a drop in performance when angle is used instead of curvature. The other 50% seems to perform as well for angle as it does for curvature. The conclusion that curvature is therefore the single most important parameter seems not to be supported by the data. Since the authors also describe W-sensitive neurons (subsection "Primary whisker neuronal activity during whisking is predicted by moment”), it seems more optimal to present the data in W-sensitive, curvature-sensitive and angle-sensitive fractions of the population data (how many neurons were recorded from in n=10 animals?).

2) Angle changes as a function of curvature as presented in Figure 4—figure supplement 1. This is very informative for the interpretation of Figure 4 and I would suggest moving Figure 4—figure supplement 1 into the main manuscript. Since angle changes dramatically during touch for individual pole positions (up to 20 degrees change in whisker angle for a fixed pole position), it can be concluded that angle is not independent from curvature and this probably underlies the range of reliability measures in curvature GLM and angle GLM. The authors should better discuss how the angle-curvature inter-dependence influences their model.

3) Passive stimulation is achieved by trimming the whiskers to 5 mm (methods). Under these conditions, it is (in my experience) impossible to induce meaningful curvature changes. The authors should better explain the experimental conditions if their experimental design allows accurate curvature measurement with a whisker trimmed to 5 mm and capillary 2 mm on whisker (Figure 4).

---

## [Author Response]

*The Reviewing editor and the reviewers agree that this is a potentially very important work that defines the features of whisking and vibrissa contact that cause a trigeminal ganglion neuron to spike. Further, while this latter topic has been explored extensively in anesthetized animals, there is limited work in the awake case. Thus a study of parameters that lead to spiking in the awake case would be a welcome contribution to the vibrissa field. Yet there are major weaknesses in the current submission that must be addressed, with additional analysis and potentially new experiments, before we can proceed further.*

Major issues:

The choice of the angular variable was absolute angle – which is likely to be a poor choice. The analysis should be redone in terms of deflection angle. The change in position of the mystacial pad also needs to be taken into account in the analysis, as noted by multiple reviewers.

As detailed in response to Reviewer 1 (point 1) and Reviewer 2, we have re-analysed the video data to extract push angle and performed new GLM analysis to consider how push angle is encoded. However, we also argue (Reviewer 1, point 1) that including analysis of absolute angle in the manuscript is useful, since it offers novel, direct evidence in favour of the view developed on behavioural grounds (Bagdasarian et al. 2013) that absolute angle only weakly influences mechanoreceptors in the follicle.

*Related to this is a need to re-evaluate claims, as noted by reviewer 3 "[…]only 50% of the data shows a drop in performance when angle is used instead of curvature. The other 50% seems to perform as well for angle as it does for curvature. The conclusion that curvature is therefore the single most important parameter seems not to be supported by the data." This requires reanalysis, yet also may require some additional data to increase the spike count.*

As detailed in response to Reviewer 3 (point 1), we have performed new statistical tests on the angle vs. curvature prediction performance, for each unit, on an individual basis. We found that the cases where angle-based GLMs performed slightly better than curvature-based GLMs could be attributed, in all but a single case, to chance. This analysis confirms our original report that curvature predicts spikes more accurately than angle.

*The application of the GLM was done with relatively few spikes, about 550 or 275 each for training and testing sets for 8 feature parameters and an unstated number of spike history parameters, which gives the editor pause. It would be thus imperative for the author to show the feature vector and the history term for at least a few units, as opposed to just predictions. In fact, the feature vector is a main point of such an analysis.*

As suggested, we now illustrate stimulus filter and spike history filters for example units in Figure 2—figure supplement 3. We agree with the editor’s concern and were at pains, in this study, to avoid over-fitting. To this end, we used regularization and cross-validation. We also kept the number of free parameters to a minimum: in most cases, 8 was the *total* number of parameters – and the Results have been revised to clarify this: “(8 total fitted parameters) and: 5 for stimulus filter, 2 for history filter, 1 bias”). In fact, when we cut down the number of free parameters even further (to 4), we found statistically indistinguishable spike prediction results (p=0.35, signed-rank test). This argues that the spike prediction performance we report is not artificially inflated by over-fitting. Thus, we are confident that we have not over-estimated the predictive power of the models.

*Further, while the recorded data show clear cycle-by-cycle whisks (Figure 1), and "whisking" cells in the trigeminal ganglion faithfully respond on a cycle-by-cycle basis, these fast changes do not appear in the predicted spike rates of Figure 2. The reason for this omission, as pointed out by Reviewer #1, needs to be explained.*

In this study, we primarily considered coding at a time-scale corresponding approximately to one whisking cycle (~100 ms); we did not consider faster time-scales. As detailed in response to Reviewer 1 (point 4): (1) focus on the ~100 ms time-scale is justified due to the relatively slow dynamics that characterize whisking against a smooth pole; (2) new analysis, where we vary the time-scale shows that our main result that curvature predicts spikes better than angle is robust across time-scales; (3) in our experiment, where we do not have the benefit of repeated trials over which to average, it is extremely difficult to test the capacity of a model to predict spikes at millisecond precision.

*Further on the topic of analysis, Reviewer #2 notes "The authors use the poor performance of angle GLMs during active pole exploration as evidence that curvature changes are what drive PWNs. But it is known that touch dominates PWN spiking responses, so including touch periods when assessing angle GLM decoding will trivially result in very low GLM angle performance. More interesting would be comparing the performance of angle GLMs during non-touch, free whisking periods with performance of curvature GLMs during touch periods." The critical issue is to determine if there is a big difference between passive and active states or if, within strong statistics, there is not a big difference. This requires reanalysis, yet also may require some additional data to increase the spike count.*

We have carried out the analysis suggested by Reviewer 2. As detailed below, the results support our original conclusion: performance of angle-based GLMs remains low, even when restricted to data from non-touch episodes. We agree that it would be interesting to compare responses in active vs. passive states but please note that it was not the aim of the current study to compare the responses to passive and active pole-contact and we do not have data that speaks to this point. To address the point would require an entirely new study. We have revised the introductory text to clarify the aims of our study.

Please read through the attached thorough and very thoughtful reviews by three of your colleagues and please address all of the issues raised in a cover letter to accompany the resubmitted manuscript.

*Reviewer #1:*

*This is an elegant work, addressing a crucial question – what do sensory neurons code during active exploration and touch – in a professional manner. The paradigm is simple and clear, and the paper is well written (for the most part), expressing clear thinking and straight-forward reasoning. This impressive work can potentially advance the understanding of sensory coding significantly. Yet, in its current form there is a danger that the paper will instead increase the confusion in the field – this is due to several major flaws that need to be carefully addressed.*

We thank the reviewer for stating that “this impressive work can potentially advance the understanding of sensory coding significantly”. We address his/her concerns below.

*1) The choice of the angular variable for analysis. The authors analyze the angle of the whisker relative to the head – let's call this the "absolute angle" here. There are 2 problems with it. One, portions of the pad rotate with significant angles during active whisking such that the absolute angle of a whisker changes but this has no effect on the shaft-follicle mechanical interactions (the entire complex moves together). This can be seen in the supplementary video of the paper, when examining the pad. Thus, the angle should be measured relative to the pad surrounding the whisker and not to the fixed head. Second, the relevant angular variable in the sensory coding game is most likely the change in angle upon contact. Both the "push angle" (Quist and Hartman, 2012; Bagdasarian et al., 2013) and the "Angle absorption" (Bagdasarian et al., 2013) carry meaningful information. This analysis of relative angular changes upon touch will also make the angle-curvature comparison more symmetric (currently the change in curvature upon touch is compared with the absolute angle).*

In response, we have conducted new analysis to consider encoding of push angle and have revised the manuscript to include it (detailed below). Our rationale for considering absolute angle in this study is that it is important for the animal to know location of its whiskers relative to the head, and hence it is interesting to assess how well absolute angle (and quantities such as amplitude, phase and set point derived from it) might be encoded. Including analysis of this quantity connects our study to a substantial previous literature (Szwed et al. 2003; Leiser and Moxon 2007; Khatri et al. 2009; Petreanu et al. 2012; Huber et al. 2012; Hires et al. 2015; Moore et al. 2015; Peron et al. 2015). Our result that Primary Whisker Neurons (PWNs) encode absolute angle poorly is fully consistent with the reviewer’s argument that absolute angle has relatively little effect on follicle mechanoreceptors. Yet, to the best of our knowledge, this point has never previously been demonstrated electrophysiologically in the awake animal. We take the point that this connection was not discussed in the original submission and have revised the Discussion to do so:

Discussion: “Our finding that whisker angle predicts PWN spikes poorly indicates that whisker angle can change without modulating mechanotransduction in the follicle. This is consistent with evidence that, during artificial whisking, the follicle-shaft complex moves as a rigid unit (Bagdasarian et al. 2013).”

To address the reviewer’s suggestion to consider push angle, we have re-analysed our video data to extract push angle during touch episodes. For each recorded unit, we trained a GLM to predict PWN spikes from push angle (using data from touch episodes). In the revised text, Figure 1 now includes push angle; and the Results have been revised to report that:

Results: “GLMs based on curvature change also predicted spike trains more accurately than GLMs based on “push angle” – the change in angle as the whisker pushes against an object (Figure 1; median PCC 0.25 IQR 0.04-0.45; p=0.006, signed-rank test). Moreover, prediction accuracy of GLMs fitted with both push angle and curvature change (median PCC 0.52, IQR 0.2-0.69) inputs was no better than that of GLMs fitted with curvature alone (p = 0.43, signed-rank test)”.

*2) After reading through the Results section it turns out that this study actually (re) revealed two types of cells – those termed by Szwed et al. (2003) as "Whisking" cells and "Touch" cells. As in Szwed et al. (2003), the former respond to whisking in air and are sensitive to the phase of whisking and the latter respond to touch and are sensitive to curvature changes. This fact should be described at the outset (Abstract, Introduction, Discussion) and compared with the relevant previous reports.*

What seems to be missing here are two complementary analyses – the sensitivity of "Whisking" cells to touch and of "Touch" cells to whisking in air. Thus, the fractions of pure whisking and touch cells, and that of a combined "whisking-Touch" type (Szwed, 2003) is not clear. True, the cell count is not high (I believe it is total of 20, although this was hard to dig – please state it at the outset) but the cost of this should not be in flattening all types to one common denominator.

In order to address this concern, we have, first, revised the text to introduce the previously demonstrated existence of subtypes of PWN cell:

Introduction: “They are both functionally and morphologically diverse; including types responsive to whisker-object contact and/or whisker self-motion (Szwed et al. 2003; Ebara et al. 2002).”

Second, we have performed new analysis to determine whether there are corresponding subtypes in our data. “Whisking sensitive” units were defined in the submitted version of the manuscript (Figure 3). In the revised manuscript, we now define “curvature-sensitive” units as those for which the curvature-based prediction of spikes is statistically significant (Figure 2 left, black circles; Peron et al. 2015). We found that:

Results: “At the level of individual units, 90% had above chance PCC and we termed these ‘curvature-sensitive’”.

Results: “Consistent with Szwed et al. (2003), PWNs were diverse: 45% were curvature-sensitive (significant PCC for curvature based GLM); 45% were both whisking and curvature sensitive and 5% were whisking sensitive but not curvature-sensitive.”

The Discussion section has been revised to consider these findings:

Discussion: “Our results also provide a mechanical basis for previous findings: our finding of subtypes of curvature-only and curvature-acceleration PWNs is consistent with previous reports of ‘touch’ and ‘whisking-touch’ units (Szwed et al. 2003; 2006).”

*Importantly, the point in the paper where the reader realizes that half of the cells are "Whisking" cells is a confusing point, reflecting back on the initial analysis. For clarity, the separation between cell types should be clarified at the beginning.*

This has been done:

Results: “As detailed below, PWNs were diverse, with some responding only to touch, others also to whisker motion”.

Concerning cell count, this is indeed 20 and this is stated in the first sentence of Results. “We recorded the activity of single PWNs from awake mice (Figure 1, Figure 1—figure supplement 1) as they actively explored a metal pole with their whiskers (N = 20 units).” For clarity, we now also report the cell count in the Materials and methods section (subsection “Electrophysiological data analysis”, first paragraph).

*3) The Abstract statement "[…]we found that primary neuron responses were poorly predicted by kinematics but well-predicted by rotational forces acting on the whisker[…]" is not supported by the data. In fact, the insisting on a single mechanical variable does not make much sense, is not convincing and, as said, is not consistent with the data presented in the paper. I strongly recommend re-considering it. First, the paper shows that half of the cells (the whisking cells) are actually sensitive to a kinematic variable, acceleration. Indeed, it is associated with force but aren't all kinematic changes associated with forces? Also, selecting whisker acceleration instead of other angular variables, such as phase and velocity, and even angle itself, seems to be arbitrary. As for the Touch cells, indeed the curvature is correlated with various angular variables, but the parameters of these correlations depend on the interactions with external objects (see Bagdasarian et al., 2013), interactions that are not investigated here. In fact, Bagdasarian et al. showed that relying on a single mechanical variable must lead to ambiguity about external features.*

We agree with the reviewer that the quoted sentence of Abstract needed reconsideration – it was misleading. The reviewer’s comment made us realize that part of the rationale for our analysis was poorly explained. Our aim was to assess the influence of two types of rotational force (moment) on PWN activity: first, the moment associated with whisking bending; second, the moment associated with whisker motion. We did this by means of proxies. We used curvature as a proxy for bending moment, since bending moment is closely related to the curvature of a whisker at its base (Birdwell et al. 2007). We used angular acceleration as a proxy for the rotational force acting on the follicle during whisker self-motion, since, through the rotational form of Newton’s second law, these two quantities are proportional, under our experimental conditions (Quist et al. 2014). For this reason, we argue that our choice to use acceleration for our analysis is not arbitrary, but well-motivated by the physics. As the reviewer points out, angular acceleration is a kinematic variable but, in the sense that it is a proxy for moment, it can also be considered a “mechanical” variable. However, we now realize that it was confusing to make a dichotomy between kinematic and mechanical. We have rewritten the text as follows.

Abstract: “Using Generalised Linear Models, we found that primary neuron responses were poorly predicted by whisker angle, but well-predicted by rotational forces acting on the whisker”.

Materials and methods: “As expressed by Newton’s second law of rotational motion, the moment – or torque –of a rigid body, rotating in a plane, is proportional to the body’s angular acceleration. […] Thus, to assess whether such a moment is encoded by PWNs, we measured angular whisker acceleration during free whisking as a proxy.“.

The reviewer’s other concern here is that “insisting on a single mechanical variable does not make much sense”. We agree with the reviewer that mechanical variables other than moment may also be encoded by the system. To address this, we also quantified both axial and lateral contact forces during episodes of whisker-pole contact. We found these variables to be strongly associated with bending moment (Figure 2—figure supplement 2) and not, therefore, under our experimental conditions, to convey independent information. It will, in future work, be interesting to explore different experimental conditions, where different mechanical variables may decouple. We have revised the Discussion to consider this issue.

Discussion: “In this study, we considered PWN encoding under conditions of pole contact, since this is well-suited to reaction force estimation […] by spike timing on a finer time-scale (Panzeri et al. 2001; Petersen et al. 2001; Arabzadeh et al. 2005; Bale et al. 2015).”

*4) The paper deals only with slow dynamics of coding – in time scales of seconds and resolution > 100 ms. Analysis at higher temporal resolutions (as was impressively done by the Petersen lab previously) is probably not possible in the current challenging setup of TG recording in awake animals. Yet, perceptual processing depends crucially on within-cycle millisecond time scales. This should be emphasized at the outset and discussed in relation to candidate sensory variables and relevant external features. It seems that while this slow time course may be relevant to features such as object radial distance (in the case of touch) and intensity of whisking (combination of whisking amplitude and frequency, which determine average acceleration throughout the cycle – see Figure 3), but not object azimuthal position, texture or shape and not phase within the whisking cycle. Also, the choice of 100 ms should be justified, and the dependency of the results on this choice should be described.*

We agree with the reviewer that coding time-scales are task-dependent and range widely. As the reviewer notes, the natural time-scale of pole exploration is slow (of order 100 ms). Thus, it is appropriate to use a corresponding time-scale for the analysis of neuronal responses in the current task. However, we take the reviewer’s point that finer time-scales of coding are likely to be important in other behavioural contexts and, as suggested, we have revised the text to discuss this:

Discussion: “In this study, we considered PWN encoding under conditions of pole contact, since this is well-suited to reaction force estimation […] by spike timing on a finer time-scale (Panzeri et al. 2001; Petersen et al. 2001; Arabzadeh et al. 2005; Bale et al. 2015).”

Furthermore, as suggested, we tested the dependency of the results on the time-scale of analysis. We did this by repeating the GLM analysis using a variety of lengths of smoothing filter (1-100ms). In all cases, curvature-based GLMs were better predictors of PWN activity than angle-based ones (signed-rank test, p<0.05, after Bonferroni correction):

Results: “The result was also robust to time-scale: prediction accuracy based on curvature was significantly greater than that based on angle for smoothing time-scales in the range 1-100ms (signed-rank test, p<0.05, Bonferroni-corrected).”

Materials and methods: “To test whether the results were robust to the smoothing time-scale, the above procedure was repeated for a range of box-car smoothing filters (1, 5, 10, 20, 50, 70 ms).”

The reviewer also makes an important, technical point. Almost all previous studies of ms-precise spike-timing (e.g., Optican and Richmond, 1987; de Ruyter et al. 1997; Reinagel and Reid, 2000; Panzeri et al. 2001; Johansson and Birzneiks, 2004; Arabzadeh et al. 2006; Montemurro et al. 2007; Gollisch and Meister, 2008; Lottem and Azouz, 2011; Bale et al. 2015) have been carried out in anaesthetized animals and have relied fundamentally on the ability to repeat the precise same mechanical stimulus many times (“trials”). In this way, one can make very precise measurements of the variability in spike timing (jitter) from trial to trial: noise is attenuated by trial-averaging. As the reviewer correctly implies, it is difficult to analyze the dynamics of coding on the ms time-scale in the awake, behaving animal. In the awake, actively whisking rodent, there are no precisely repeated trials and trial-averaging is not an available option. As we show in Figure 2—figure supplement 1, the lack of repeated trials was a limiting factor for model prediction accuracy in our study. Our strategy was to take advantage of the natural, ~100 ms time-scale of whisker-object interactions in our pole exploration paradigm, to perform averaging in the temporal domain.

*5) Figure 1 show a whisker that pushes against the object during retraction. The video and Figure 1 shows the "standard" contact, during protraction. The authors should make it clear whether their analysis was based on both directions. If so, this comment becomes a major one – the authors must include the direction as one of the analyzed variables and describe the dependency of the various findings on it.*

The reviewer is correct that our analysis is based on both directions. Our GLM framework encompasses directional selectivity as follows. In a curvature GLM, the preferred direction is determined by the sign of the stimulus filter. Positive stimulus filter coefficients tend to make positive curvature change trigger spikes; negative coefficients make negative curvature trigger spikes. The directional selectivity of a GLM is particularly simple when the stimulus filter is a single number. This is relevant to our study, since we found that such simple GLMs predicted spikes as well as GLMs with a length 5 stimulus filter (Figure 2—figure supplement 1). When a whisker pushes against an object during protraction, curvature increases; when it pushes against an object during retraction, it decreases. Materials and methods have been revised:

Materials and methods: “Since we found that GLM performance was just as good with *L_k_*= 1 as *L_k_* = 5 (Figure 2—figure supplement 1), we used […] when it pushes against an object during retraction, it decreases.”

To address the reviewer’s point, we conducted further analysis, using the above approach, to determine the directional selectivity of every curvature-sensitive unit:

Results: “At the level of individual units, 90% had above chance PCC and we termed these ‘curvature-sensitive’ (Materials and methods). Of the curvature-sensitive units, 61% were sensitive to positive curvature change and 39% to negative curvature change (Materials and methods).”

*Also, curvatures are very strong in this study (Figure 2, movie). Please refer to it and compare to free-head conditions in which often the minimal impingement principle (Prescott et al., 2013) applies. Please discuss the implications on the predominance of curvature coding in this study.*

The maximum curvatures in our study (~0.2 mm^-1^) are consistent with previous studies of object localisation and tactile maze navigation in head-fixed mice where curvature has been measured (O’Connor et al. 2010; Sofroniew et al. 2014). Unfortunately, to the best of our knowledge, there are no published studies of free-head mice that report whisker curvature, which makes it difficult to compare our data to free-head conditions. Nevertheless, if we understand correctly, the reviewer’s concern is with the sensitivity of PWNs to smaller curvatures. We can address this point, since our dataset contains a wide range of curvature change values, by constructing tuning curves. We have added example tuning curves to Figure 2—figure supplement 1 which show that neuronal firing rate was modulated not only by strong curvature change (~0.2 mm^-1^), but also weaker ones (~0.01 mm^-1^), associated with subtler pole contact. Also, the new curvature graph in the movie shows spikes evoked by small curvature change.

*Reviewer #2:*

Key findings:

*1) PWNs are relatively insensitive to absolute whisker angle but highly sensitive to curvature change.*

*2) The degree to which PWNs are tuned to curvature change predicts their response to inertial force during free whisking.*

These results are well supported by the data, and the data is valuable, nicely collected and presented. However, the results don't change the general understanding of PWN coding and thus are not novel.

We thank the reviewer for saying that “the data is valuable” and that our main findings are “well supported by the data”, but we respectfully disagree with the statement that our results are not novel. We agree that our results do not falsify the Hartmann model of PWN coding but we argue that, given the necessary simplifications in any theoretical model, and the daunting complexity of the awake, behaving situation, it is remarkable to find clear support for the theory. As we discuss in the manuscript (subsection “Mechanical framework for tactile coding”, first paragraph), Hartmann and colleagues (Solomon and Hartmann 2006; Birdwell et al. 2007) proposed that the evidence that PWNs respond to touch (Szwed et al. 2003; 2006) could, in principle, be explained by the hypothesis that mechanoreceptors in the follicle are sensitive to whisker bending moment. However, before our work, no study had ever directly tested the theory by simultaneously measuring both PWN activity and whisker reaction forces. This is significant, since the theory makes simplifying approximations and assumptions (for example, concerning boundary conditions) that it is unsafe to assume will necessarily hold in the awake, behaving animal. Since the whisker system is an important model in neuroscience, we argue that this was a major gap in the literature and that our study makes a substantial contribution to putting our general understanding of somatosensory coding on a more solid, mechanical basis.

*The paper focuses on overturning a straw-man characterization of the literature, that PWNs are tuned to absolute whisker angle, not deflection forces.*

*It is unfair to characterize the current results as "at odds with passive stimulation studies (Gibson 1983, Lichtenstein 1990[…])". The classic studies refer to PWN tuning to angle of deflection not absolute angle. These particular studies had no ability to assess PWN tuning in the absence of deflection. In Bale (2013), again the positional tuning was in the context of positional deflection not free whisking angle. Indeed, Leiser (2007) showed that firing rates are 10x higher in PWNs during contact than during awake free whisking. The logical interpretation of this and many other cited studies of PWN coding is that deflection-induced forces (often quantified as deflection angle) are the primary driver of PWN spiking, not whisker position absent deflection.*

We agree with the reviewer that passive whisker stimulation can be understood from a force-encoding point of view and we accept that the quoted sentence from the original submission (“at odds with[…]”) lacked nuance. We have revised the text to remove the sentence (Introduction, last paragraph).

Our contention is that there is an apparent contrast (not, we agree, a contradiction) between passive stimulation studies and our data, and that our study sheds light on why this is so. With passive stimulation, whisker angle (more precisely, change in whisker angle with respect to a whisker’s resting angle) correlates with firing rate (Zucker and Welker 1969; Gibson and Welker 1983; Lichtenstein et al. 1990; Jones et al. 2004; Arabzadeh et al. 2005; Bale and Petersen 2009; Lottem and Azouz 2011; Bale et al. 2013) whereas, in our data, whisker angle did not predict PWN firing rate. Thus, apparently the same variable exerts different effects under the two conditions. The reason that this is not actually a contradiction (consistent with the reviewer’s remark) is, as we show, that passive stimulation does not just change whisker angle but also bends the whisker (Figure 4 and Figure 4—figure supplement 2). This characteristic has, however, not been widely appreciated: passive stimulation is usually carried out on shortened whiskers, which are so stiff that the bending is hard to detect unless, as here, measured with high-resolution imaging.

*The authors use the poor performance of angle GLMs during active pole exploration as evidence that curvature changes are what drive PWNs. But it is known that touch dominates PWN spiking responses, so including touch periods when assessing angle GLM decoding will trivially result in very low GLM angle performance. More interesting would be comparing the performance of angle GLMs during non-touch, free whisking periods with performance of curvature GLMs during touch periods.*

Our primary aim was to seek input variables that could predict spikes during pole exploration as a whole – and we therefore argue that it is important to test the GLM’s performance on long episodes encompassing both contact and non-contact. However, the comparison suggested by the reviewer adds insight and we thank him/her for the suggestion. We performed new analysis and found that:

Results: “Curvature GLMs also predicted spikes during touch episodes significantly more accurately (median PCC 0.57, IQR 0.23-0.72) than did angle GLMs during non-touch episodes (median 0.06, IQR 0.02-0.35; p=0.005, signed-rank test)”.

*In the study, active touch occurs at multiple pole positions, while passive deflections have only one starting position. Thus the comparison of curvature and angle coupling between active and passive conditions (Figure 4) is apples to oranges. For example, if the mouse must position his whisker 10 degrees more protracted to contact the pole in one position vs. another during active sensing, the correlation between angle and curvature will be degraded when averaged across pole positions. Including non-touch periods in the analysis further degrades the correlation. Thus the poor cross-correlation for the awake condition in Figure 4 is trivial.*

We agree that the weak cross-correlation between angle and curvature change for the awake condition are explicable in the terms stated by the reviewer and shown by us in Figure 4—figure supplement 1 of the original submission. However, we respectfully disagree that this makes these results “trivial” and note that other reviewers found these results insightful and requested that they be moved into the main text (new Figure 4).

*The more interesting and fair comparison is the extent to which active control of whisker position impacts the relationship between curvature and push angle. Push angle is defined as the angle through which the whisker is rotated into the object (see Quist and Hartmann, 2012 or Hires, 2013 for details). Active control could alter the rigidity of the follicle, impacting follicle stresses and thus spiking activity of PWNs. This should be detectable via comparing the difference in push angle/curvature coupling (i.e. the slope of touch trajectories in 4E, assuming curvature was measured at the same radial distance) between active and passive states.*

We thank the reviewer for the suggestion to consider push angle. In response, we have re-analysed all our video data to extract push angle during touch episodes. Using these data, we have performed new GLM analysis to consider the relation between push angle and PWN spiking in the awake state. For each recorded unit, we trained GLM to predict PWN spikes from push angle. We have revised Figure 1 to show push angle and Results:

Results: “GLMs based on curvature change also predicted spike trains more accurately than GLMs based on “push angle” […] inputs was no better than that of GLMs fitted with curvature alone (p = 0.43, signed-rank test).”

*Additional comments:*

*The data in the paper are interesting and do have potential to address some open questions that would increase the importance and novelty of the work. Some possible ideas that reanalysis could address, (in order of increasing interest):*

*1) Do PWNs that are tuned to acceleration direction show the same directional selectivity to deflection direction?*

We have performed new analyses to address this point (see Materials and methods, subsection “Responses to whisking without touch”, last paragraph and subsection “Generalised Linear Model (GLM)”, first paragraph). For the whisking sensitive units, reported in Figure 3, we found that:

Results: “For 70% of whisking-sensitive units, directional selectivity for acceleration was consistent with that for curvature.”

*2) Do force components (Faxial, bending moment) differentially drive PWNs?*

We agree that this is an interesting possibility. However, under our experimental conditions, we found that axial force is strongly (nonlinearly) associated with bending moment (Figure 2—figure supplement 2). The relationships so strong (under our conditions) that we were concerned that it is difficult to separate out contributions to PWN firing, and that it is misleading to make any strong claim about lack of differential drive. For the reviewer’s information, the performance (PCC) of axial force quadratic GLMs was median 0.35 (IQR 0.08-0.52), significantly lower than the PCC for curvature change (signed-rank test, p = 0.01). To resolve this question would require an entirely new study, investigating other stimuli.

3) Do PWN responses to passive vs. active touch exhibit different sensitivity to deflection angle or whisker curvature change?

We agree that this is an interesting question: however, it is beyond the scope of the present investigation and requires a completely new study.

*Reviewer #3:*

*The manuscript of Campagner et al. investigates the whisker parameters (angle and curvature) that allow reliable prediction of spiking of primary whisker neurons upon passive or active touch. The manuscript is potentially interesting, although I have some concern about experimental setup and the validity of comparisons between passive and active conditions. Additionally, even though curvature reliably predicts spiking in awake rats for a subset of the data, the range of quantified reliability is large and not discussed.*

*1) The major conclusion (curvature much better predicts spiking than angle) is based predominantly on Figure 2. The full range of reliability measures for curvature is 0.1 – 0.9. The authors put a lot of emphasis on the fraction of high values (max 0.88), but completely ignore the lower measures. Vice versa, the high values for angle GLMs are only briefly mentioned and emphasis put on poor predicting values. It seems very relevant to discuss the entire range for both conditions.*

We agree with the reviewer’s concern to clearly report variability in the data and, to this end, the relevant figures (Figure 2 and Figure 3) show not only averages but also all individual data points for every unit. However, we accept that we did not, in the original text, emphasize the variability. In response, we have revised the text to do so. We have also carried out new analysis detailed below.

Results: “Although the activity of most units was better predicted by whisker curvature change than by whisker angle, *there was significant variability in prediction performance, and*there were a few units for which the angle prediction performance was appreciable (Figure 2).”

Discussion:”In this study, we considered PWN encoding under conditions […] performance varied across units, suggesting that other force components may also be encoded.”

Additionally, only 50% of the data shows a drop in performance when angle is used instead of curvature. The other 50% seems to perform as well for angle as it does for curvature. The conclusion that curvature is therefore the single most important parameter seems not to be supported by the data.

We take the reviewer’s point that, for some units, the angle performance (PCC) is higher than the curvature PCC and thank him/her for the opportunity to drill deeper into this point. In principle, these might be genuine differences; alternatively, they might be statistical fluctuations in the measurements due to chance. To address this, we performed a new, cell-by-cell analysis, where we used a resampling technique (detailed, subsection “Generalised Linear Model (GLM)”, last paragraph) to statistically compare angle vs. curvature PCC for each unit, on an individual basis. The new results support the original conclusion that curvature is encoded better than angle:

Results: “Moreover, on a unit-by-unit basis, for 65% of units, curvature change GLMs predicted spikes better than angle (signed-rank test, p<0.05, Bonferroni corrected); only for 5% of units did angle predict spikes better than curvature change.”

*Since the authors also describe W-sensitive neurons (subsection "Primary whisker neuronal activity during whisking is predicted by moment”), it seems more optimal to present the data in W-sensitive, curvature-sensitive and angle-sensitive fractions of the population data.*

We have performed a new analysis to classify the recorded units as curvature and/or whisking sensitive (see also response to Reviewer #1, point 2) and report the results in the revised text:

Results: “Consistent with Szwed et al. (2003), PWNs were diverse: 45% were curvature-sensitive (significant PCC for curvature based GLM); 45% were both whisking and curvature sensitive and 5% were whisking sensitive but not curvature-sensitive.”

*(How many neurons were recorded from in n=10 animals?).*

N=20 units were recorded and this is stated in the first sentence of Results. “We recorded the activity of single PWNs from awake mice (Figure 1, Figure 1—figure supplement 1) as they actively explored a metal pole with their whiskers (N = 20 units).” For clarity, we now also report the N number in Materials and methods (subsection “Electrophysiological data analysis”, first paragraph).

*2) Angle changes as a function of curvature as presented in Figure 4—figure supplement 1. This is very informative for the interpretation of Figure 4 and I would suggest moving Figure 4—figure supplement 1 into the main manuscript.*

This has been done. These plots are now in the main text (Figure 4 of the revised manuscript).

*Since angle changes dramatically during touch for individual pole positions (up to 20 degrees change in whisker angle for a fixed pole position), it can be concluded that angle is not independent from curvature and this probably underlies the range of reliability measures in curvature GLM and angle GLM. The authors should better discuss how the angle-curvature inter-dependence influences their model.*

We take the point that angle-curvature correlation was not mentioned in the Discussion section and have revised the text to do so:

Discussion: “A challenge of studying neural coding under unconstrained, awake conditions is that sensory variables tend to correlate. A useful feature of the GLM training procedure is that it takes such correlations into account. We found that, although whisker angle predicted spikes for a subset of units, this effect was very largely explained by a curvature-coding model, together with the correlation between angle and curvature.”

*3) Passive stimulation is achieved by trimming the whiskers to 5 mm (methods). Under these conditions, it is (in my experience) impossible to induce meaningful curvature changes. The authors should better explain the experimental conditions if their experimental design allows accurate curvature measurement with a whisker trimmed to 5 mm and capillary 2 mm on whisker (Figure 4).*

In response, we include (Figure 4—figure supplement 2) video stills and raw whisker tracking data below, which we hope convinces the reviewer that measurement of curvature changes under passive stimulation is possible. The curvature changes are, of course, small and there are two main reasons why we had sufficient sensitivity to reliably measure them: (1) we study mouse not rat; mouse whiskers being thinner and less stiff, a given stimulus produces more whisker bending than it would in rat; (2) our whisker tracker achieves accurate solutions by fitting quadratic curves to the base region of the whisker (thus keeping the number of free parameters small, whilst being justified by thin beam physics; Birdwell et al. 2007; Quist and Hartman 2012; Bale et al. 2015) and takes advantage of temporal contiguity to constrain the fit by the solution in the previous frame. Since short whiskers are stiff and therefore have high moment of inertia, even small curvature changes can correspond to substantial reaction forces.

Figure 5 (top four panels) shows four video frames taken during trapezoidal, passive whisker stimulation. Whisker tracker solutions are overlaid. Curvature change (lower left) and corresponding tracker solutions (lower right) are shown for a 45 ms episode, with coloured dots marking the times of the four example frames, and shading from blue to aqua indicating time. This whisker has negative intrinsic curvature. As the actuator applies force to the whisker, the whisker straightens up and the curvature increases.

Author response image 1.**DOI:**
http://dx.doi.org/10.7554/eLife.10696.016